# Neural stem cell-encoded temporal patterning delineates an early window of malignant susceptibility in *Drosophila*

Karine Narbonne-Reveau[1†], Elodie Lanet[1†], Caroline Dillard[1†], Sophie Foppolo[1], Ching-Huan Chen[2], Hugues Parrinello[3], Stéphanie Rialle[3], Nicholas S Sokol[2], Cédric Maurange[1*]

[1]Aix Marseille Univ, CNRS, IBDM, Marseille, France; [2]Department of Biology, Indiana University, Bloomington, United States; [3]MGX-Montpellier GenomiX, Institut de Génomique Fonctionnelle, Montpellier, France

**Abstract** Pediatric neural tumors are often initiated during early development and can undergo very rapid transformation. However, the molecular basis of this early malignant susceptibility remains unknown. During *Drosophila* development, neural stem cells (NSCs) divide asymmetrically and generate intermediate progenitors that rapidly differentiate in neurons. Upon gene inactivation, these progeny can dedifferentiate and generate malignant tumors. Here, we find that intermediate progenitors are prone to malignancy only when born during an early window of development while expressing the transcription factor Chinmo, and the mRNA-binding proteins Imp/IGF2BP and Lin-28. These genes compose an oncogenic module that is coopted upon dedifferentiation of early-born intermediate progenitors to drive unlimited tumor growth. In late larvae, temporal transcription factor progression in NSCs silences the module, thereby limiting mitotic potential and terminating the window of malignant susceptibility. Thus, this study identifies the gene regulatory network that confers malignant potential to neural tumors with early developmental origins.

*For correspondence: cedric.maurange@univ-amu.fr

[†]These authors contributed equally to this work

Competing interests: The authors declare that no competing interests exist.

## Introduction

Many pediatric tumors are thought to initiate during prenatal stages and are able to rapidly progress towards malignancy, sometimes within a few months (*Marshall et al., 2014*). Yet, they contain very few genetic alterations (*Huether et al., 2014*; *Parsons et al., 2011*; *Pugh et al., 2013*; *Vogelstein et al., 2013*; *Wu et al., 2014*) suggesting that transformation in infancy is not driven by the gradual accumulation of genetic lesions over many years, as for most adult cancers. Instead, cells born during early development appear predisposed to malignant transformation. However, the developmental programs and gene networks that govern this early malignant susceptibility remain to be deciphered (*Chen et al., 2015b*).

*Drosophila* is a well-established animal model to investigate basic principles of tumorigenesis in the developing or ageing organism (*Gonzalez, 2013*; *Siudeja et al., 2015*). In particular, it has been used to demonstrate that single gene inactivation perturbing the asymmetric divisions of neural stem cells (NSCs), called neuroblasts (NBs) in *Drosophila*, during development can rapidly cause NB amplification and aggressive malignant tumors in transplantation assays (*Caussinus and Gonzalez, 2005*; *Knoblich, 2010*). However, the underlying mechanisms of transformation are still unknown (*Caussinus and Gonzalez, 2005*; *Knoblich, 2010*).

Normal NBs are active from embryogenesis to pupal stages and generate the neurons and glial cells that constitute the *Drosophila* central nervous system (CNS). Two main types of NBs have been

**eLife digest** Some aggressive brain tumors that affect children start to form before the child is even born. These tumors often develop much more rapidly than tumors found in adults, and require fewer genetic mutations to become dangerous and invasive. However, it is not known why this happens.

Fruit flies are often used as animal models for cancer studies. As the fly brain develops, cells called neural stem cells divide several times, each time producing one stem cell and another cell known as the intermediate progenitor. The intermediate progenitor can itself divide one more time before maturing to become a neuron. Different types of neurons form in different stages of brain development. This is due to the sequential production of proteins called transcription factors in neural stem cells. Each transcription factor is inherited by a different set of intermediate progenitors and alters the activity of certain genes to determine the type of neuron the cells become.

Some genetic mutations can prevent intermediate progenitors from maturing and cause them to revert to a stem-cell-like state, which allows them to rapidly divide and form tumors. Here, Narbonne-Reveau, Lanet, Dillard et al. use fruit flies to investigate why tumors that form early on in development progress so rapidly. The experiments uncover a 'molecular clock' in the neural stem cells that marks out a window of time in which they generate intermediate progenitors that are prone to becoming cancerous. This clock is represented by the sequential production of transcription factors that, in addition to determining neuronal identity, also turn off various growth-promoting genes in cells as brain development proceeds. These genes sustain normal cell division, but are silenced later on to prevent cells from dividing too many times.

If the maturation of intermediate progenitors is disrupted early on in brain development while the growth-promoting genes are still active, the molecular clock fails to switch off the growth-promoting genes. As a result, these cells acquire an unlimited ability to divide, which drives tumor growth. However, later in development when the growth-promoting genes have already been switched off, disrupting the maturation of intermediate progenitors does not lead to these cells becoming cancerous.

Therefore, Narbonne-Reveau, Lanet, Dillard et al.'s findings explain why intermediate progenitors that mature early on in brain development are more prone to becoming cancerous than those that mature later, and why they need fewer mutations to become invasive. Most of the genes involved in this process are also found in humans. Therefore, the same mechanism might govern how aggressive childhood brain tumors are, which is a question for future studies to address.

identified. Upon asymmetric division, most NBs (type-I) self-renew while giving rise to an intermediate progenitor, called the ganglion mother cell (GMC), which usually divides once to generate two post-mitotic neurons or glia. In contrast, a small number of NBs (type-II) located in the central brain region of the CNS, generates intermediate neural progenitors (INPs) that can produce a few GMCs allowing for an amplification of post-mitotic progeny in the lineage (*Homem and Knoblich, 2012*) (*Figure 1—figure supplement 1A*). NBs undergo a limited number of divisions during development and invariably stop dividing before adulthood (*Truman and Bate, 1988*). For NBs located in the ventral nerve cord (VNC) of the CNS, this limited mitotic potential is governed by a NB-intrinsic clock that schedules their terminal differentiation during metamorphosis (*Maurange et al., 2008*). This timing mechanism is set in NBs by the sequential expression of a series of 'temporal' transcription factors that has the ability to endow each progeny with a different neuronal identity according to their birth order (*Kohwi and Doe, 2013*; *Maurange, 2012*). In addition, NBs in the VNC need to progress up to a late temporal factor in the series to become competent to respond to the hormonal cues promoting cell cycle exit and terminal differentiation during metamorphosis (*Homem et al., 2014*; *Maurange et al., 2008*). In VNC NBs, there are four known temporal transcription factors (Hunchback (Hb) -> Kruppel (Kr) -> Pdm -> Castor (Cas)) mainly expressed during embryogenesis (*Baumgardt et al., 2009*; *Grosskortenhaus et al., 2005*; *Isshiki et al., 2001*; *Kambadur et al., 1998*). Cas is re-expressed in early larval NBs presumably followed by other, yet unknown, temporal factors required to set up a late global transition of neuronal identity during larval development and

to schedule NB termination during metamorphosis (*Maurange et al., 2008*). Progression throughout the sequence is governed by cross-regulatory transcriptional interactions between the temporal transcription factors, and can be blocked by continuous mis-expression of a temporal factor or by its inactivation (*Figure 1—figure supplement 1B*) (*Isshiki et al., 2001*). Transitions between temporal transcription factors can also be promoted by Seven-up (Svp), an orphan nuclear receptor orthologous to mammalian COUP-TF transcription factors. In particular, Svp is transiently expressed in embryonic NBs, to promote the early Hb->Kr transition, and in larval NBs to trigger a global temporal transition allowing NBs to switch from generating an early subpopulation of neurons expressing the BTB transcription factor Chinmo to a later sub-population expressing other markers (*Benito-Sipos et al., 2011*; *Kanai et al., 2005*; *Maurange et al., 2008*; *Mettler et al., 2006*). Inactivation of Svp during early larval stages blocks NBs in an early temporal identity. Consequently, late $svp^{-/-}$ NBs continuously generate Chinmo$^+$ neurons, fail to undergo terminal differentiation during metamorphosis, and continue to divide in adults (*Maurange et al., 2008*). Multiple series of temporal transcription factors have been uncovered in the different regions of the CNS, and recent data suggests that this temporal patterning system is evolutionary conserved and operating in mammalian NSCs (*Brand and Livesey, 2011*; *Konstantinides et al., 2015*; *Li et al., 2013*; *Mattar et al., 2015*).

Remarkably, inactivation of genes involved in the differentiation of INPs or GMCs can cause their reversion to a NB-like progenitor that, unlike normal NBs, possesses an unrestrained mitotic potential causing malignant tumors. This highly penetrant phenotype has, for example, been observed in the case of mutations inactivating the transcription factor Prospero (Pros) in GMCs (*Betschinger et al., 2006*; *Choksi et al., 2006*), or inactivating the NHL translational repressor Brat, the transcription factor Earmuff/dFezf, or components of the SWI/SNF complex in INPs (*Figure 1—figure supplement 1A*) (*Bello et al., 2006*; *Betschinger et al., 2006*; *Eroglu et al., 2014*; *Koe et al., 2014*; *Lee et al., 2006*; *Weng et al., 2010*). More recently, it has been described that inactivation of the transcription factors Nerfin1/INSM1 or Lola in post-mitotic neurons is sufficient to induce their progressive dedifferentiation into GMC- and NB-like states, and to cause unlimited proliferation (*Froldi et al., 2015*; *Southall et al., 2014*) (*Figure 1—figure supplement 1A*). While the mechanisms by which these factors induce or maintain differentiation has been thoroughly investigated and explain the observed amplification of NB-like cells upon loss-of-function, the reasons why dedifferentiated NBs (dNBs) acquire an unlimited proliferation potential remain unknown.

Here, we test the hypothesis that the unlimited mitotic potential underlying the malignant properties of dNBs is caused by the deregulation of the temporal specification system. We find that neural tumors only undergo malignant transformation if dedifferentiation is induced during an early developmental window. This early malignant susceptibility of neural cells is governed by the temporal patterning system that regulates whether or not a pre-existing early oncogenic module, that controls NB mitotic activity during development, can be co-opted to trigger malignant growth. This work therefore uncovers how the temporal transcription factor series regulates NSC mitotic potential during development and governs the malignant susceptibility of neural cells according to their birth-order. Our study provides a model that may help understand the ontogeny of human tumors with early developmental origins.

## Results

### Malignant tumors are propagated by a subset of dNBs that resists differentiation during metamorphosis

In order to precisely track the growth of single tumors throughout developmental and adult stages, we used the $pox^n$-*GAL4* driver to express GFP (*UAS-GFP*) and an RNAi construct against *pros* (*UAS-pros$^{RNAi}$*) in a targeted subset of six VNC NBs from the beginning of larval development (*Figure 1A, B* and *Figure 1—figure supplement 2*). Consequently, many dNBs (Mira$^+$) are generated at the expense of neurons (Elav$^+$), forming six tumor-like structures of proliferating progenitors in late larval VNCs (*Figure 1B* and *Figure 1—figure supplement 2*). While wild-type (*wt*) NBs undergo terminal differentiation during metamorphosis and are absent in adults (*Figure 1A*), tumors of dNBs persist in adults and continue growing, forming in 6 day-old adults, large tumors that have fused and invaded the whole VNC (*Figure 1B–D*). At this stage, tracking of $pox^n$>*pros$^{RNAi}$* tumors generated in the VNCs shows that they invade adjacent tissues such as the central brain or halters (*Figure 1C*).

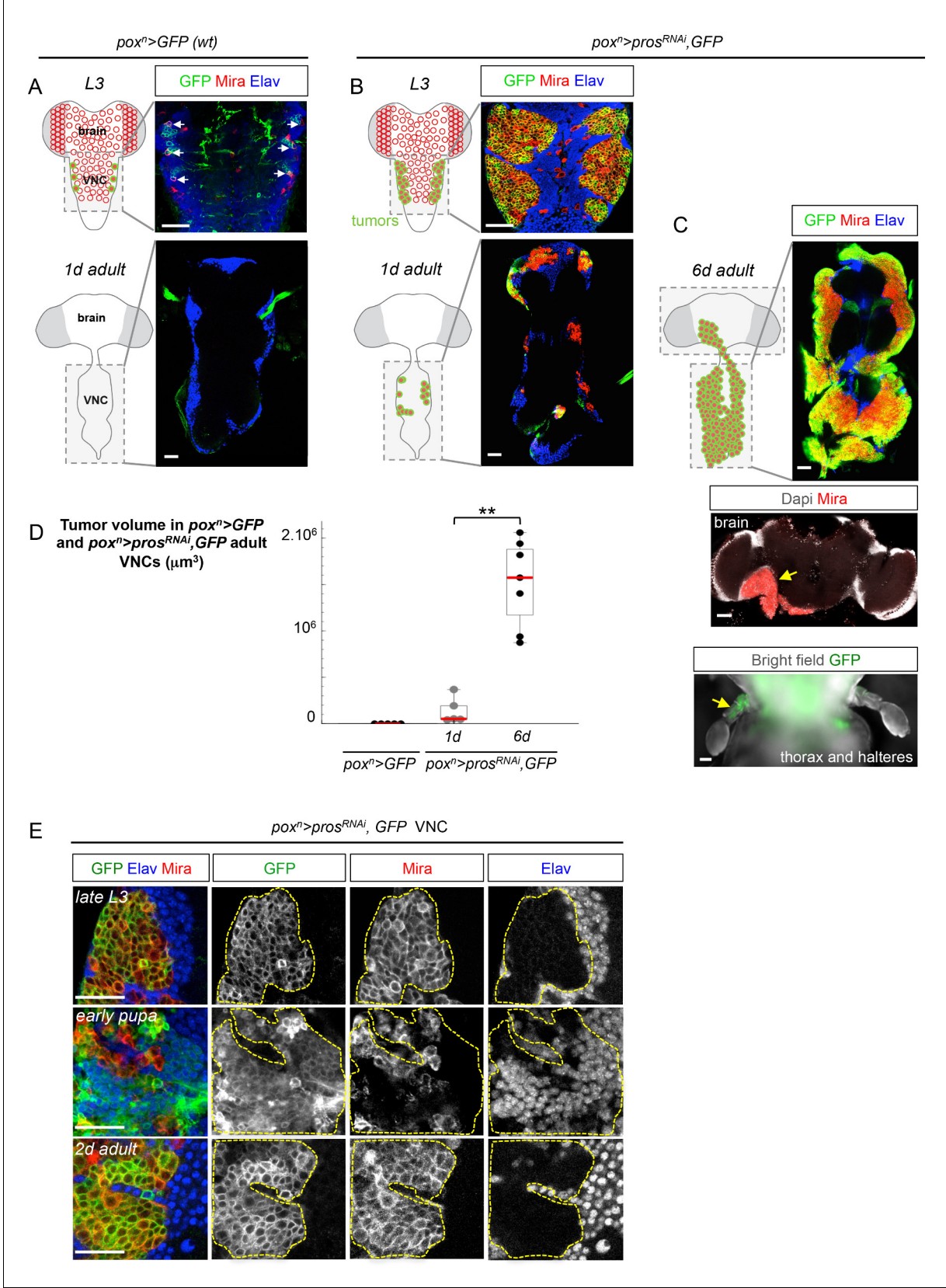

**Figure 1.** A subset of dNBs induced by Pros knock-down propagates malignant tumors in adults. The scale bar in all images represents 30 μm. NBs and dNBs are always labeled using an anti-Mira antibody. Neurons are labelled using anti-Elav. (**A**) Schematic drawing representing a ventral view of
*Figure 1 continued on next page*

*Figure 1 continued*

the late larval (*L3*) and adult *Drosophila* CNS. Ventral nerve cord (VNC). NBs are represented as red circles. The *pox^n-GAL4* driver is active in six lateral NBs of the larval VNC (marked in green on the scheme). In *pox^n-GAL4, UAS-GFP* larvae (*pox^n>GFP*), GFP labels the six NBs (white arrows) and their recently generated progeny due to transient GAL4 and GFP perdurance. All NBs are absent in the adult VNC. (B) In *pox^n-GAL4, UAS-pros^RNAi, UAS-GFP, UAS-dcr2* larvae (*pox^n>pros^RNAi, GFP*), six tumors of dNBs are generated. dNBs are represented on the scheme as green circles filled in red. A subset of dNBs persist and form small tumors in 1 day-old adult VNCs. (C) In 6 day-old adults, *pox^n>pros^RNAi, GFP* tumors cover the whole VNC and invade adjacent tissues such as the brain, and halteres (D) Mean tumor volumes quantified in *wt pox^n> GFP* adult VNCs and in *pox^n>pros^RNAi, GFP* 1 and 6 day-old adult VNCs. No tumor is observed in *wt* adults. 1 day-old *pox^n>pros^RNAi, GFP* VNCs (n= 5 VNCs, m = 1.4x10$^5$, SEM = 6.3x10$^4$) and 6 day-old *pox^n>pros^RNAi, GFP* VNCs (n = 7 VNCs, m = 1.5x10$^6$, SEM = 1.8x10$^5$). p-value is 2.5x10$^{-3}$. (E) *pox^n>pros^RNAi, GFP* tumors are almost exclusively composed of dNBs in late L3, and devoid of neurons (Elav). At around 20 hr after pupa formation, a brief pulse of neuronal differentiation in *pox^n>pros^RNAi, GFP* tumors is seen. GFP briefly labels recently differentiated Elav$^+$ neurons due to transient GAL4 perdurance. In adults, persisting dNBs reconstitute malignant tumors.

The following figure supplements are available for figure 1:

**Figure supplement 1.** Progeny-to-NSC dedifferentiation and temporal progression in the developingcentral nervous system of *Drosophila*.

**Figure supplement 2.** *pox^n>pros^RNAi* larvae possess tumors in the VNC but not in the brain.

**Figure supplement 3.** L1/L2-induced MARCM *pros^-/-* clones generate malignant tumors in adult.

---

Therefore, NB tumors induced by the loss of Pros during early larval stages and maintained in their natural environment resist differentiation cues operating during metamorphosis, and invariably acquire an unlimited growth potential as well as invasive properties. As such, we define them as malignant tumors.

Tracking *pox^n>pros^RNAi* tumors throughout development revealed that a large number of dNBs undergo neuronal differentiation at around 20 hr after pupa formation (APF). At this time, a large population of GFP$^+$ neurons (up to 79% of the tumor cell population ( ± 3%; n = 3416; 3 tumors)) could be transiently observed due to the transitory persistence of GFP from dNBs (*Figure 1E*). Similar figures of neural differentiation were not observed in tumors when examined during larval or adult stages. To ensure that this burst of differentiation was not inherent to the use of RNAi, nor specific to the *pox^n* lineages, we used the MARCM technology (*Lee and Luo, 1999*) to generate random *pros^-/-* clones during early larval development (L1/L2). While such clones lead to malignant tumor growth in adults (*Figure 1—figure supplement 3A*), a similar burst of differentiation was observed in most lineages at around 20 hr of metamorphosis (*Figure 1—figure supplement 3B*). Interestingly, this event coincides with the timing of *wt* NB terminal differentiation that occurs in response to the production of the steroid hormone (*Homem et al., 2014*; *Maurange et al., 2008*). Thus, most dNBs retain the competency to undergo differentiation like *wt* NBs during metamorphosis. In contrast, a subset of differentiation-resisting dNBs persists in adult to propagate malignant tumors.

## A subset of dNBs aberrantly maintains the early transcription factor Chinmo

Unlimited tumor growth could be propagated by a resetting of the temporal series in newly-born dNBs. However, of the four known temporal factors in VNC NBs (Hb -> Kr -> Pdm -> Cas) and Svp, only Cas was occasionally detected in tumors observed in late larvae and in adults (*Figure 2—figure supplement 1*). Nevertheless, removing ectopic Cas from *pros^RNAi* tumors did not affect the ability to generate large tumors in adults (*Figure 2—figure supplement 2*). Therefore, the persistence of proliferating tumors in adults is neither caused by a reset nor by a stalling at an early stage of the temporal series in dNBs. We then concentrated on Chinmo, a transcription factor known to label a sub-population of early-born neurons in type-I lineages in larvae (*Maurange et al., 2008*; *Zhu et al., 2006*), (*Figure 2A*). We find that Chinmo is not only expressed in early-born neurons of type-I and type-II lineages (*Figure 2—figure supplement 3*) but also in early NBs from which they are generated. In the VNC, Chinmo is highly expressed in early larval NBs (L1 and L2) and their progeny, but its expression in NBs progressively decreases from early L3 and is switched off in most NBs and their subsequent progeny by midL3 (*Figure 2A* and *Figure 2—figure supplement 4*). Moreover, when

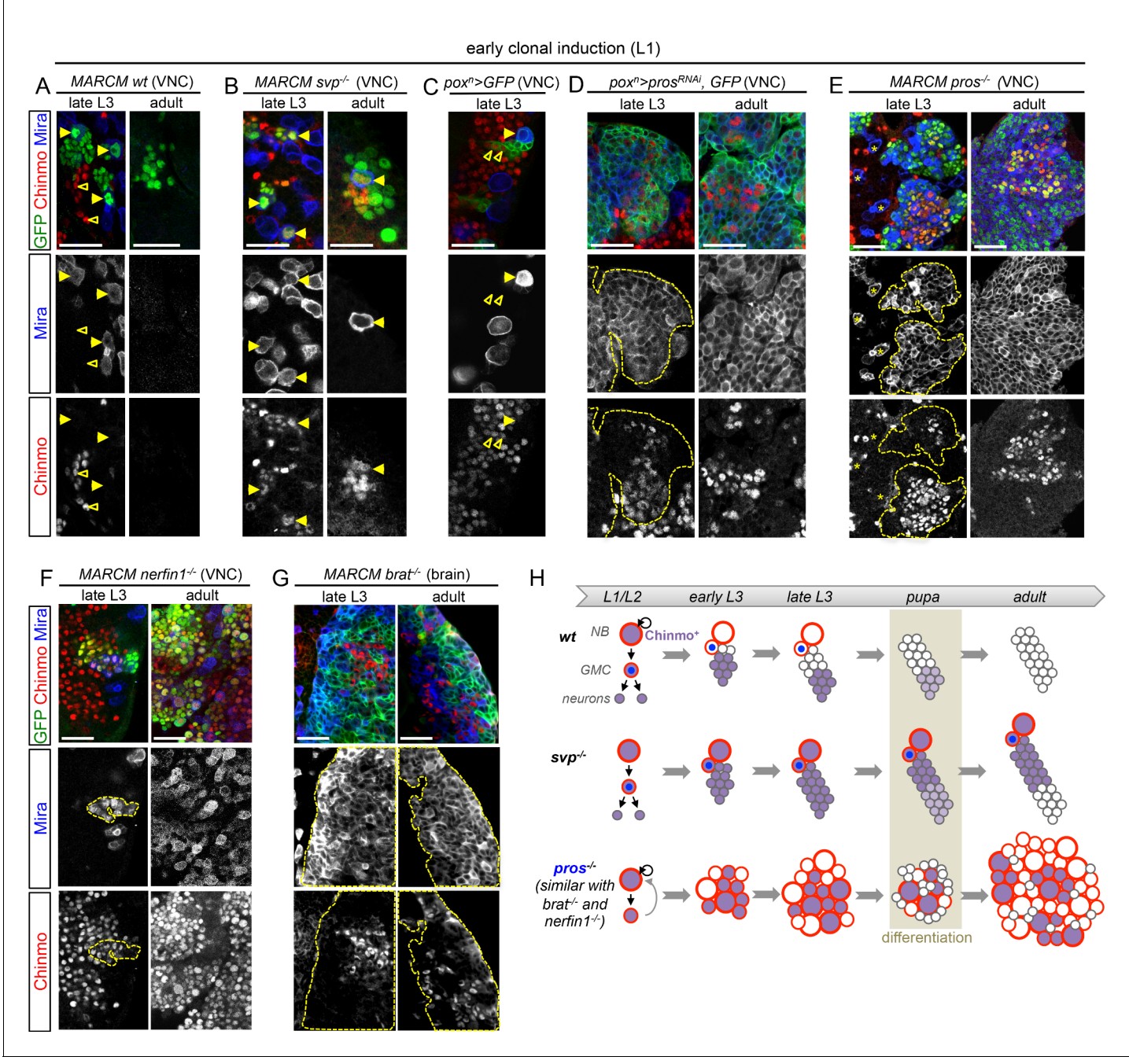

**Figure 2.** Chinmo is ectopically expressed in tumors induced by dedifferentiation. All clones are induced in L1 (24 hr after larval hatching) using MARCM and labeled with GFP. (**A**) NBs in *wt* clones (arrows) have silenced Chinmo at late L3 stages. Note that at this stage, Chinmo remains strongly expressed in early born neurons (empty arrowheads). In adults, NBs are absents in *wt* clones. Chinmo is not expressed anymore in early-born neurons in adults. (**B**) NBs (arrows) in *svp*[−/−] clones maintain Chinmo in late L3. Chinmo is also maintained in *svp*[−/−] NBs persisting in adults and their newly generated neurons (arrow). (**C**) *pox*[n] NBs in late L3 have silenced Chinmo (arrows) while Chinmo expression is observed in early-born neurons (empty arrowheads). (**D**) A subset of *pox*[n]>*pros*[RNAi] dNBs maintains Chinmo in late L3 and adults. (**E**) A subset of dNBs in VNC *pros*[−/−] MARCM clones maintains Chinmo at late L3 stages. All surrounding *wt* NBs have silenced Chinmo (asterisks). Aberrant Chinmo expression is maintained in a subset of dNBs in adult *pros*[−/−] clones. (**F**) A subset of dNBs in *nerfin*[−/−] clones maintains Chinmo in late L3 and adult VNCs. (**G**) A subset of dNBs induced in *brat*[−/−] clones maintains Chinmo in late L3 and adult brains. (**H**) During early development (from L1 to mid-L3), Chinmo (purple) is expressed in NBs and early-born neurons. It is silenced in NBs during mid-larval stages by the progression of the temporal series. In *svp*[−/−] mutant NBs, Chinmo is maintained in NBs and their progeny up to adulthood. In *pros*[−/−], *nerfin*[−/−]or *brat*[−/−] tumors, Chinmo escapes temporal regulation in a subpopulation of dNBs and remains expressed in tumors as development progresses.

*Figure 2 continued on next page*

*Figure 2 continued*

The following figure supplements are available for figure 2:

**Figure supplement 1.** The temporal factors Hb, Kr and Pdm and the Svp nuclear receptor are not expressed in larval *pros*$^{RNAi}$ tumors.
**Figure supplement 2.** Ectopic expression of the temporal factor Cas in early-induced *pros*$^{-/-}$ tumors does not contribute to persistence in adult.
**Figure supplement 3.** Chinmo is a marker of early-born neurons in type-II lineages.
**Figure supplement 4.** Chinmo is expressed in young but not old NBs.
**Figure supplement 5.** Percentage of Chinmo-expressing dNBs in tumors.

the late L2 pulse of Svp in NBs was abrogated, by inducing *svp*$^{-/-}$ MARCM clones in L1, in order to block temporal patterning progression, we observed that NBs failed to silence Chinmo in L3 and adults (*Figure 2B*). Therefore, Chinmo is a marker of early temporal identity and its window of expression in NBs is terminated in early L3 by progression of the temporal transcription factor series (*Figure 2H*).

Interestingly, while Chinmo is silenced in *pox*$^n$ NBs in late L3 (*Figure 2C*), we observed that a minor subset of dNBs in *pox*$^n$>*pros*$^{RNAi}$ tumors retained Chinmo expression (*Figure 2D*). Ectopic expression of Chinmo in about 20% of dNBs was also observed in tumors that persist in adults (*Figure 2D* and *Figure 2—figure supplement 5*). Moreover, aberrant expression of Chinmo in dNBs is not specific to the *pox*$^n$ lineage as it can be observed in L3 and adult *pros*$^{-/-}$ MARCM clones induced in L1 from random NBs (*Figure 2E*). Thus, in *pros*$^{-/-}$ tumors induced during early larval stages, *chinmo* in a subset of dNBs appears to escape regulation by the temporal series and fails to be silenced at the appropriate time (*Figure 2H*). Similar aberrant temporal expression of Chinmo was also observed in tumors originating from the dedifferentiation of immature neurons in *nerfin1*$^{-/-}$ MARCM clones (*Figure 2F*) and in tumors originating from the dedifferentiation of INPs in *brat*$^{-/-}$ MARCM clones from type-II NB lineages (*Figure 2G*). Thus, ectopic Chinmo expression appears to be a common feature of neural tumors induced during early larval stages by dedifferentiation (*Figure 2H*).

## Chinmo promotes resistance to differentiation and is necessary to sustain tumor growth

To test whether Chinmo's aberrant expression in dNBs contributes to tumorigenesis, we co-expressed *chinmo*$^{RNAi}$ and *pros*$^{RNAi}$ transgenes in random NB clones induced during early larval stages, or from different GAL4 drivers active in specific NB subsets. In all cases, despite efficient *chinmo* silencing (*Figure 3—figure supplement 1A*), the presence of supernumerary Mira$^+$ cells found dividing in late L3 larvae demonstrated that Chinmo was dispensable for dedifferentiation and initial NB amplification upon loss of Pros (*Figure 3A,B* and *Figure 3—figure supplement 1B,C*). However, we find that *pros*$^{RNAi}$, *chinmo*$^{RNAi}$ tumors are smaller than *pros*$^{RNAi}$ tumors in late L3 larvae (*Figure 3A,B*, *Figure 3—figure supplement 1C,D*), due to smaller and fewer dNBs that exhibited a decreased mitotic index (*Figure 3D*, *Figure 3—figure supplement 1E–G*). In contrast, forced expression of Chinmo (using *UAS-chinmo*) in a *pros*$^{RNAi}$ context enhanced the size of tumors in late L3 larvae, which contained larger dNBs (*Figure 3—figure supplement 1D,E*). These results show that Chinmo promotes cell growth and proliferation within tumors.

Importantly, a higher proportion of dNBs is converted to neurons in *pros*$^{RNAi}$, *chinmo*$^{RNAi}$ tumors compared to *pros*$^{RNAi}$ tumors during metamorphosis (*Figure 3—figure supplement 1B,H*). Consistently, adult persistence of tumors originating from *pros*$^{RNAi}$, *chinmo*$^{RNAi}$ clones induced in L1/L2 is reduced by about 30% compared to *pros*$^{RNAi}$ clones (*Figure 3—figure supplement 2A*). This indicates that Chinmo confers resistance to the differentiation cues operating during metamorphosis. Additionally, persisting *pros*$^{RNAi}$; *chinmo*$^{RNAi}$ tumors are smaller and fail to grow further, or completely differentiate, in 6 day-old adults, while *pros*$^{RNAi}$ control clones rapidly invade the CNS (*Figure 3A–C* and *Figure 3—figure supplement 2B*). Failure to grow is, at least, partly due to a low mitotic activity within *pros*$^{RNAi}$; *chinmo*$^{RNAi}$ adult tumors, as already observed in larval tumors.

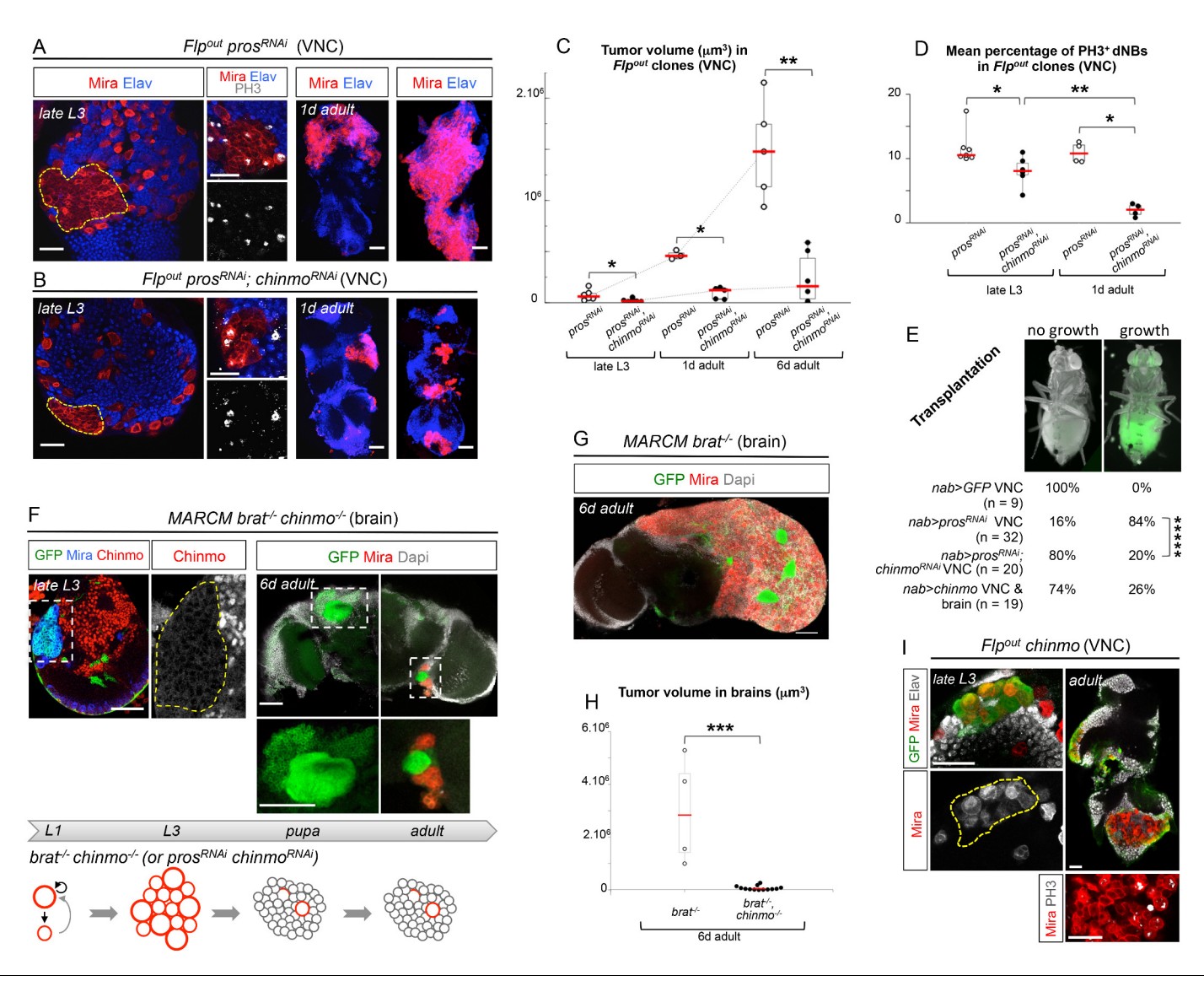

**Figure 3.** Chinmo sustains tumor growth beyond developmental stages. All clones were induced in L1/L2. (**A**) Expression of *pros^RNAi* in Flp-out clones induces malignant tumors, covering the VNC in adults. (**B**) Expression of both *pros^RNAi* and *chinmo^RNAi* in Flp-out clones induces tumors that fail to grow further in the adult VNC. (**C**) Mean tumor volume in Flp-out *pros^RNAi* and *pros^RNAi;chinmo^RNAi* clones induced during early larval stages quantified in the VNC of wandering L3 (*wL3*), 1 day-old and 6 day-old adults. wL3: *pros^RNAi* (n = 6 VNCs, m = 7.4x10^4, SEM=2.1x10^4), *pros^RNAi;chinmo^RNAi* (n = 7 VNCs, m = 2.3x10^4, SEM = 5.7x10^3); 1 day-old adult: *pros^RNAi* (n = 3 VNCs, m = 4.7x10^5, SEM = 2.5x10^4), *pros^RNAi;chinmo^RNAi* (n = 5 VNCs, m = 9.5x10^4, SEM = 2.5x10^4); 6 day-old adult: *pros^RNAi* (n = 5 VNCs, m = 1.5x10^6, SEM = 2.1x10^5), *pros^RNAi;chinmo^RNAi* (n = 6 VNCs, m = 2.4x10^5, SEM = 1.9x10^5). p-values are respectively 2.2x10^-2, 3.6x10^-2 and 8.7x10^-3. (**D**) Mean percentage of PH3+ dNBs in late L3 and 1 day-old adult Flp-out *pros^RNAi* and *pros^RNAi;chinmo^RNAi* induced during early larval stages. Late L3: *pros^RNAi* (n = 7 VNCs, m = 11.64, SEM=0.99), *pros^RNAi;chinmo^RNAi* (n = 6 VNCs, m = 8.06, SEM = 0.92); 1 day-old adult: *pros^RNAi* (n = 4 VNCs, m = 10.92, SEM = 0.79), *pros^RNAi;chinmo^RNAi* (n = 4 VNCs, m = 1.97, SEM = 0.50). p-values are respectively 1.4x10^-2 and 2.9x10^-2; p-value between *pros^RNAi;chinmo^RNAi* wL3 and 1-day old adults is 9.5x10^-3. (**E**) Tumorigenic growth after transplantation of VNCs is assessed by the presence of GFP in the abdomen of transplanted flies after 7 days (p-*value is 6.0x10^-6*). (**F**) MARCM *brat^-/-*, *chinmo^-/-* clones induced during early larval stages generate tumors (Mira) in late L3. However, most clones undergo complete neuronal differentiation during metamorphosis, as shown with an absence of dNBs and large ectopic axonal bundles in adult clones (inset). Occasional remaining dNBs are not able to reconstitute large tumors (inset). Below, *brat^-/-*, *chinmo^-/-* clones are represented schematically during development. (**G**) MARCM *brat^-/-* clones induced during early larval stages rapidly leads to large malignant tumors in the adult brain. (**H**) Mean tumor volumes in MARCM *brat^-/-* and *brat^-/-*, *chinmo^-/-* clones induced during early larval stages quantified in 6 day-old adult central brains. MARCM *brat^-/-* clones (n = 4 brains, m = 3.0x10^6, SEM = 1.0x10^6) and MARCM *brat^-/-*, *chinmo^-/-* clones (n = 13 brains, m = 7.1x10^4, SEM = 3.3x10^4). p-value is 8.4x10^-4. (**I**) Overexpression of *chinmo* in Flp-out clones induces NB amplification in larvae (yellow dotted line), giving rise to tumors composed of proliferating dNBs in adults.

*Figure 3 continued on next page*

*Figure 3 continued*

The following figure supplements are available for figure 3:

**Figure supplement 1.** Chinmo knock-down leads to reduced tumor growth and increased differentiation.

**Figure supplement 2.** *pros*^RNAi^ *chinmo*^RNAi^ tumors fail to become malignant in adults.

**Figure supplement 3.** Chinmo⁺ dNBs exhibit a higher mitotic index than Chinmo⁻ dNBs within *pros*^RNAi^ tumors.

Moreover, the mitotic rate in *pros*^RNAi^; *chinmo*^RNAi^ tumors decreases from larval to adult stages, while remaining stable in *pros*^RNAi^ tumors (*Figure 3D*), in 6 day-old adults, *pros*^RNAi^; *chinmo*^RNAi^ dNBs sometimes show cytoplasmic extensions characteristic of quiescence (*Chell and Brand, 2010*; *Truman and Bate, 1988*) (*Figure 3—figure supplement 2C*). Together, these experiments suggest that dNBs in *pros*^RNAi^; *chinmo*^RNAi^ tumors progressively exhaust their proliferation potential. The decreased growth potential of *pros*^RNAi^; *chinmo*^RNAi^ tumors was confirmed with the conventional transplantation assay (*Rossi and Gonzalez, 2015*) with only 16% (n=20) of VNCs developing large tumors in the abdomen of transplanted animals after 7 days, compared to 84% (n=32) with *pros*^RNAi^ VNCs (*Figure 3E*). We also assessed the mitotic index of the Chinmo⁺ and Chinmo⁻ dNBs present within the same *pros*^RNAi^ tumor. We found that it was significantly higher for the subpopulation of Chinmo⁺ dNBs than for Chinmo⁻ dNBs suggesting that two types of progenitors with different mitotic potential co-exist in tumors (*Figure 3—figure supplement 3*).

We then tested whether Chinmo was also essential for the growth of *brat*^-/-^ tumors. Like *brat*^-/-^ MARCM clones (*Figure 2G*), *brat*^-/-^ *chinmo*^-/-^ MARCM clones induced in L1 led to large clones with amplified NBs in late L3 (*Figure 3F*). However, while *brat*^-/-^ MARCM clones led to large tumors partly covering the adult brain in 6 day-old adults (*Figure 3G*), *brat*^-/-^ *chinmo*^-/-^ MARCM clones differentiated during metamorphosis and occasional remaining dNBs were unable to reconstitute large tumors in adults (*Figure 3F,H*). Thus, these results collectively indicate that ectopic Chinmo in a subset of dNBs boosts cell growth, counteracts neuronal differentiation during pupal stages, and is required for sustained proliferation once development is terminated.

To further investigate Chinmo's ability to promote cell growth and proliferation on its own, we over-expressed it in *wt* NBs clones from early larval stages. Although Chinmo does not induce proliferation when expressed in post-mitotic neurons (*Zhu et al., 2006*), we found that clonal overexpression of *UAS-chinmo* in *wt* NBs and GMCs from early L1 is sufficient to induce NB amplification in 60% of NB clones (6 VNCs, 37 clones) and the formation of tumors that resist differentiation during metamorphosis and continue proliferating in adults (*Figure 3I*). Because adult flies containing *UAS-chinmo* clones rapidly die, we could not test the long-term growth potential of *UAS-chinmo* NB tumors. To circumvent this problem, we transplanted larval VNCs over-expressing Chinmo in all NBs (*nab>chinmo*). We found that 26% of transplanted flies grow large tumors in the abdomen after 7 days (*Figure 3E*) showing that Chinmo over-expression in NBs and their GMCs is sufficient to induce sustained tumorigenic growth. Together, these data demonstrate that aberrant Chinmo expression in a subset of dNBs is oncogenic and drives sustained tumor growth beyond developmental stages.

## Chinmo boosts protein biosynthesis and expression of the mRNA-binding proteins Imp and Lin-28

To explore Chinmo's mode of action, we compared, by RNA-seq, the transcription profiles of late larval VNCs in which all NBs express the *pros*^RNAi^ construct (*nab>pros*^RNAi^) with *pros*^RNAi^ VNCs in which Chinmo was either knocked down or over-expressed (respectively *nab>pros*^RNAi^, *chinmo*^RNAi^ and *nab>pros*^RNAi^, *chinmo*) (*Figure 4A*). Two hundred and fourteen genes were both up-regulated in the *nab>pros*^RNAi^, *chinmo* condition and down-regulated in the *nab>pros*^RNAi^, *chinmo*^RNAi^ condition when compared to the *nab>pros*^RNAi^ control (p-value < 0.05). They were considered as putative targets positively regulated by Chinmo. On the other hand, 388 genes were both downregulated in the *nab>pros*^RNAi^, *chinmo* condition and up-regulated in the *nab>pros*^RNAi^, *chinmo*^RNAi^ condition compared to the *nab>pros*^RNAi^ control (p-value<0,05) (*Figure 4—source data 1*). They were considered as putative targets negatively regulated by the presence of Chinmo. Gene ontology (GO) and KEGG pathway analyses indicate that negative targets are highly enriched in

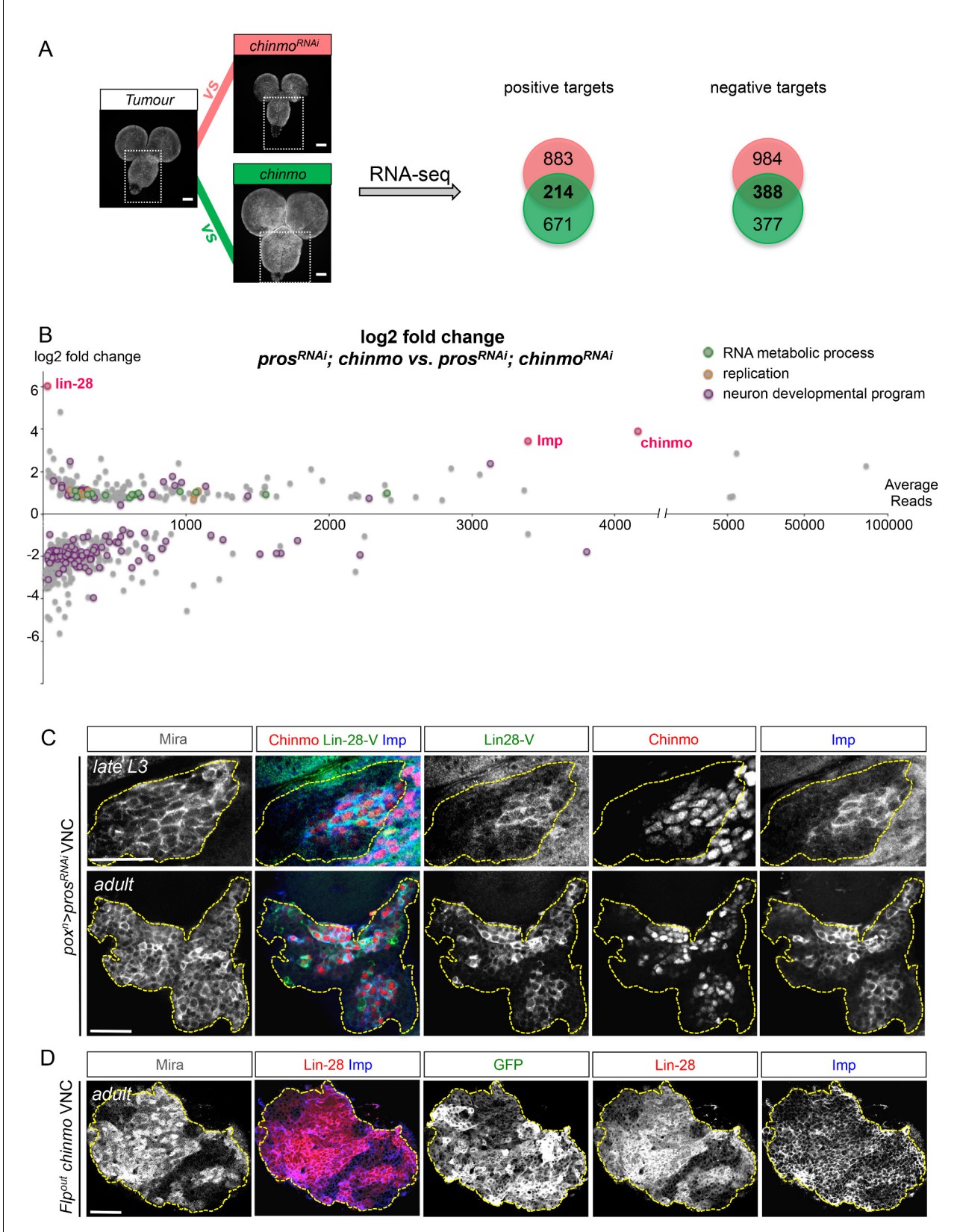

**Figure 4.** Chinmo promotes Imp and Lin-28 expression. (**A**) *pros^RNAi* is expressed in all larval NBs using *nab*-GAL4. RNA-seq indicated 214 genes to be commonly up-regulated, and 388 genes were found to be commonly downregulated, when comparing *nab>pros^RNAi vs. nab>pros^RNAi, chinmo^RNAi* and
*Figure 4 continued on next page*

*Figure 4 continued*

*nab>pros*<sup>RNAi</sup>, *chinmo* vs. *nab>pros*<sup>RNAi</sup> (adj p-value < 0.05). (**B**) Graphical representation of the log2 fold change as a function of the base mean expression of Chinmo targets, comparing *nab>pros*<sup>RNAi</sup>, *chinmo* to *nab>pros*<sup>RNAi</sup>, *chinmo*<sup>RNAi</sup>. lin-28, Imp and chinmo are highlighted in red. (**C**) Lin-28, Imp (cytoplasmic) and Chinmo (nuclear) are co-expressed in the same subset of dNBs in *pox*<sup>n</sup>*>pros*<sup>RNAi</sup> larval and adult tumors (delineated with the yellow dashed lines). (**D**) Clonal mis-expression of Chinmo in GFP⁺ Flp-out clones induced in L1, delimited by yellow dashed lines, induces Imp and Lin-28 co-expression in dNBs.

The following source data and figure supplements are available for figure 4:

**Source data 1.** Differentially expressed genes and enriched GO terms and KEGG pathways between *pros*<sup>RNAi</sup> tumors expressing various levels of Chinmo.
**Figure supplement 1.** Transcriptional analysis summary.
**Figure supplement 2.** Chinmo is necessary for Imp expression in tumors. (A,B) Imp is expressed in *pros*<sup>RNAi</sup> and *brat*<sup>-/-</sup> tumors induced in L1/L2.

genes involved in neuronal differentiation consistent with the ability of Chinmo to prevent dNB differentiation during metamorphosis (*Figure 4B* and *Figure 4—figure supplement 1*). In contrast, positive targets are enriched in genes involved in ribosome biogenesis, RNA and DNA metabolism, DNA replication and cell-cycle progression (*Figure 4B* and *Figure 4—figure supplement 1*). These data support a role for Chinmo in promoting dNB growth and mitotic activity that is consistent with our genetic and proliferation assays (*Figure 3*).

Among the 214 positive targets of Chinmo uncovered by RNA-seq, several genes are important regulators of malignancy in mammals such as *lin-28* (*Molenaar et al., 2012*), *IGF-II mRNA-binding protein (Imp)* (*Lederer et al., 2014*), *musashi (msi)* (*Wang et al., 2010*), *Aldehyde dehydrogenase (Aldh)* (*Ginestier et al., 2007*), and *snail (sna)* (*Barrallo-Gimeno and Nieto, 2005*) (*Figure 4—figure supplement 1*, *Figure 4—source data 1*). Among them, the most highly up-regulated genes, when comparing the *nab>pros*<sup>RNAi</sup>, *chinmo* vs. *nab>pros*<sup>RNAi</sup>, *chinmo*<sup>RNAi</sup> conditions, are two mRNA-binding proteins Lin-28 (64 fold) and Imp (10 fold) (*Figure 4B*, *Figure 4—figure supplement 1*, *Figure 4—source data 1*). Lin-28 and Imp are highly conserved oncofoetal genes in humans (*Bell et al., 2013*; *Lederer et al., 2014*) and have recently been shown to be co-expressed in embryonic NSCs in mice (*Yang et al., 2015*). We validated their expression in tumors by immunostaining. Interestingly, in larval and adult *pros*<sup>RNAi</sup> tumors, both Imp and Lin-28 are present in the cytoplasm of the subset of dNBs that co-express Chinmo (*Figure 4C*). Furthermore, clonal overexpression of *UAS-chinmo* induces supernumerary NBs that express both Imp and Lin-28, while small *pros*<sup>RNAi</sup>, *chinmo*<sup>RNAi</sup> or *brat*<sup>-/-</sup> *chinmo*<sup>-/-</sup> tumors in adults lack Imp (not tested for Lin-28) (*Figure 4D* and *Figure 4—figure supplement 2*). Together, these experiments demonstrate that Imp and Lin-28 are, direct or indirect, positive targets of Chinmo in tumors.

## Imp sustains Chinmo expression in dNBs and tumor growth

In mammals, the three orthologs of Imp (IMP1-3, also called IGF2BP1-3) are believed to be important regulators of tumorigenesis, but their function and targets are unclear (*Bell et al., 2013*). We find that overexpression of Imp in NBs is not sufficient to induce their amplification and tumorigenesis (*Figure 5—figure supplement 1*). Moreover, efficient Imp knock-down using one or two different *Imp*<sup>RNAi</sup> transgenes (*Imp*<sup>RNAi1</sup> or *Imp*<sup>RNAi1;2</sup>) in *pox*<sup>n</sup>*>pros*<sup>RNAi</sup> larvae did not prevent initial NB amplification, although tumor growth appeared slowed (*Figure 5A,B* and *Figure 5—figure supplement 2A,B*). Strikingly, most *pros*<sup>RNAi</sup>, *Imp*<sup>RNAi</sup> tumors failed to maintain continuous growth and remained small in 6 day-old adults (*Figure 5B–C*). Thus Imp is required for sustained tumor growth.

A recent study has shown that Imp post-transcriptionally promotes Chinmo expression in mushroom body neurons (*Liu et al., 2015*). We thus sought to assess Chinmo expression in *pros*<sup>RNAi</sup> tumors lacking Imp. We find that Chinmo⁺ dNBs are still present in larvae. However, they are absent from many tumors in 6 day-old adults (12 out of 22 with *Imp*<sup>RNAi1</sup>, and 11 out of 14 with *Imp*<sup>RNAi1;2</sup>) (*Figure 5A,B*). Similar results are obtained with tumors grown for 10 days in larvae fed with a sterol-free diet that has been shown to prevent pupariation (*Katsuyama and Paro, 2013*; *Parkin and Burnet, 1986*). In such conditions, *pox*<sup>n</sup>*>pros*<sup>RNAi</sup> tumors grow extensively and invade the central brain and optic lobes, while *pox*<sup>n</sup>*>pros*<sup>RNAi</sup>, *Imp*<sup>RNAi1;2</sup> tumors remain much smaller, stay

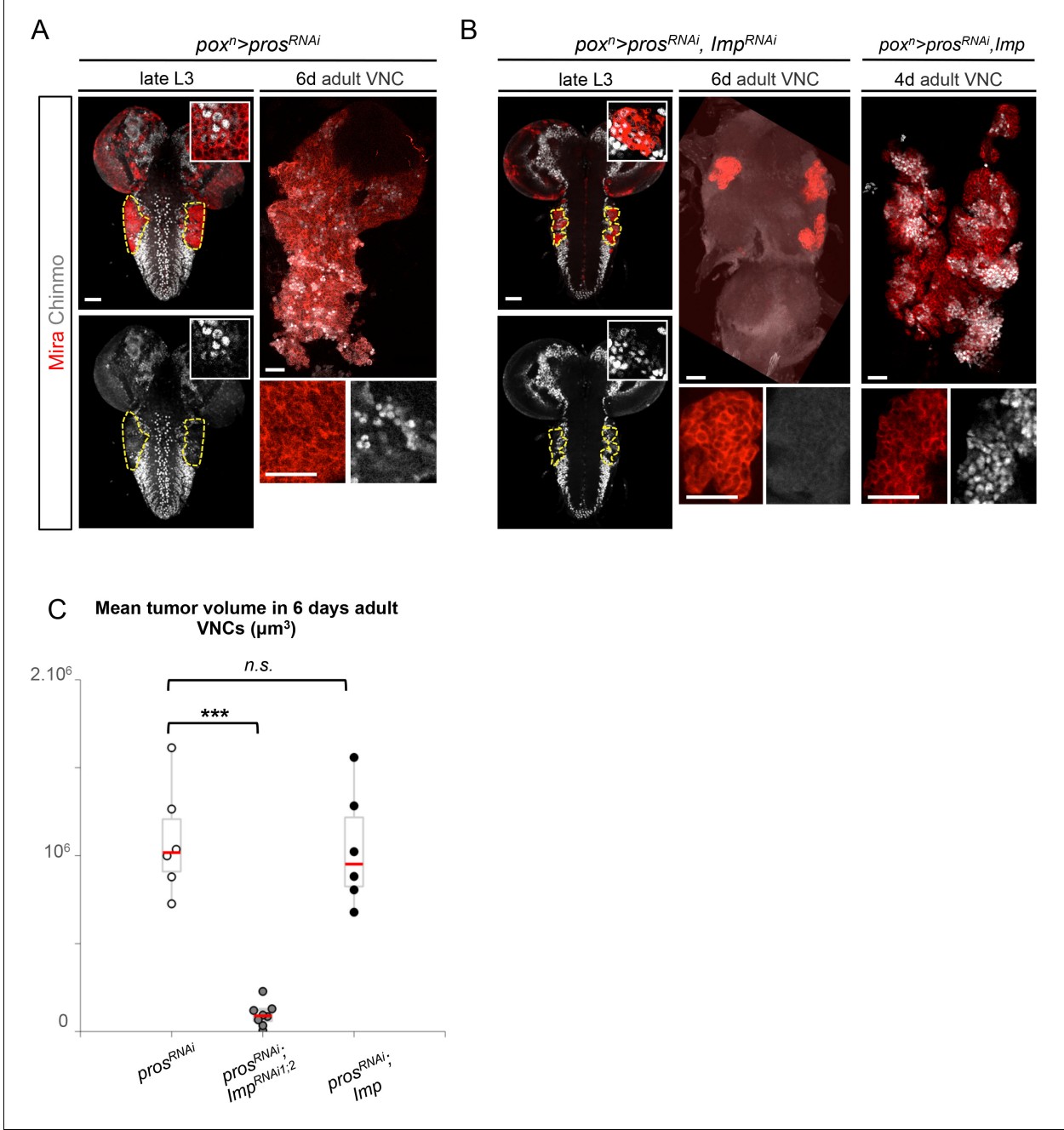

**Figure 5.** Imp sustains Chinmo expression in tumors. (**A**) Chinmo is expressed in a subset of dNBs in both larval (dashed yellow lines) and adult $pox^n>pros^{RNAi}$ tumors (see enlargement). (**B**) Chinmo is still expressed in a subset of dNBs in larval $pox^n>pros^{RNAi}$, $Imp^{RNAi}$ tumors but is progressively lost in adult tumors that remain small (see enlargement). $pox^n>pros^{RNAi}$, $Imp$ tumors tend to have an increased number of Chinmo+ cells (not quantified). (**C**) Mean tumor volume of 6 day-old adults in $pox^n>pros^{RNAi}$, $pox^n>pros^{RNAi}$; $Imp^{RNAi1;2}$ and $pox^n>pros^{RNAi}$; $Imp$. $pox^n>pros^{RNAi}$ (n = 6 VNCs, m = $1.1 \times 10^6$, SEM = $1.3 \times 10^5$), $pox^n>pros^{RNAi}$; $Imp^{RNAi1;2}$ (n = 8 VNCs, m = $9.5 \times 10^4$, SEM = $2.4 \times 10^4$), $pox^n>pros^{RNAi}$; $Imp$ (n = 6 VNCs, m = $1.0 \times 10^6$, SEM = $1.3 \times 10^5$). p-values are respectively $6.7 \times 10^{-4}$ and 0.82.

The following figure supplements are available for figure 5:

**Figure supplement 1.** Imp mis-expression is not sufficient to initiate tumors.

**Figure supplement 2.** Imp knock-down decreases tumor growth.

**Figure supplement 3.** Imp is necessary to sustain tumor growth and Chinmo expression.

localized to the VNC and show an almost complete loss of Chinmo$^+$ dNBs (*Figure 5—figure supplement 3A,B*). We also noted a tendency for an increased number of Chinmo$^+$ dNBs in 4 day-old adults upon Imp over-expression (*Figure 5B*, not quantified). These experiments indicate that Imp is not required to establish the initial population of Chinmo$^+$ dNBs but is necessary for their long-term maintenance in tumors. Together with the transcriptional activation of Imp by Chinmo, these experiments reveal a positive feedback loop between Chinmo and Imp necessary for sustained tumor growth beyond developmental stages.

## Lin-28 boosts Chinmo expression within tumors

In contrast to Imp, loss of Lin-28 does not appear to impair the self-renewal of Chinmo$^+$ dNBs and the growth of *pros$^{-/-}$* or *brat$^{-/-}$* tumors induced in L1 (tumor size quantified in adults) (*Figure 6—figure supplement 1*). This suggests that Lin-28 is dispensable for Chinmo expression and for sustained tumor growth. However, while mis-expression of *Drosophila lin-28* in *wt* larval NB clones is not sufficient to induce NB amplification and ectopic expression of Chinmo and Imp (*Figure 6—figure supplement 2*), we find that overexpression of *Drosophila lin-28* in *pros$^{RNAi}$* tumors from their initiation leads to a significant increase in the proportion of Chinmo$^+$ dNBs and an overall increase in the intensity of Chinmo expression when observed in 6 day-old adults (*Figure 6A–C*; *Figure 6—figure supplement 3*). Moreover, Imp is co-expressed in all Chinmo$^+$ dNBs. This suggests that overexpression of Lin-28 in the tumorigenic context is able to favor the self-renewing capacity of Chinmo$^+$/Imp$^+$ dNBs at the expense of Chinmo$^-$/Imp$^-$ dNBs. Interestingly, overexpression of mammalian LIN28A and LIN28B is often associated with malignant tumors in human (*Carmel-Gross et al., 2016*; *Molenaar et al., 2012*; *Viswanathan et al., 2009*). Remarkably, human LIN28A and LIN28B, mis-expressed in *Drosophila pox$^n$>pros$^{RNAi}$* tumors, have retained the ability to strongly enhance Chinmo expression within tumors, demonstrating evolutionary-conserved interactions between Lin-28 and Chinmo or its regulators (*Figure 6A–C*). All together, these data identifies Chinmo and Imp as a core oncogenic module that sustains dNB proliferation and tumor growth beyond developmental stages. In addition, expression of this module is boosted by high levels of Lin-28 in tumors (*Figure 6D*, *Figure 6—figure supplement 4*).

## The temporal series silences *chinmo*, *Imp* and *lin-28* (CIL genes) in late NBs to limit their mitotic potential

We then wondered whether Chinmo, Imp and Lin-28 (CIL) co-expression is specific to dNBs or can be found during development. Interestingly, in the VNC and central brain, both Imp and Lin-28 are co-expressed with Chinmo in normal NBs and their surrounding neuronal progeny from L1 to early L3. From early L3, they are then progressively downregulated together with Chinmo in NBs and subsequent progeny (*Figure 7A*). Lin-28 and Chinmo remain transiently expressed in early-born neurons up to midL3 and early pupal stages respectively, while Imp expression in early-born neurons perdures in adult (*Figure 7—figure supplement 1*). However, *Imp* and *lin-28*, like *chinmo*, fail to be silenced in late larval and adult NBs with a stalled temporal series (*svp$^{-/-}$*), as well as in their progeny (*Figure 7B*). Thus, the temporal system silences *chinmo*, *Imp* and *lin-28* in L3 NBs. We then tested the function of Chinmo in NBs during development. *chinmo$^{-/-}$* MARCM clones were induced during embryogenesis in order to ensure the removal of Chinmo during the entire post-embryonic period (*Figure 7C*). In *chinmo$^{-/-}$* late L3 clones, we did not observe a difference in clone size in the VNC compared to the control *wt* clones (*Figure 7D*) and *chinmo$^{-/-}$* NBs exhibit the same mitotic rate and possess the same size as *wt* NBs in late L3 (*Figure 7E,F*). Thus Chinmo is dispensable for NB growth and proliferation during larval development. We then tested whether temporally blocked NBs persist in adults due to the maintenance of Chinmo, Imp and Lin-28. Knocking-down Chinmo in temporally stalled *svp$^{-/-}$* NBs is not sufficient to prevent adult persistence, although NB growth is impaired (*Figure 7G,H* and *Figure 7—figure supplement 2A*). However, knocking down both Chinmo and Imp in *svp$^{-/-}$* NBs restores their elimination before adulthood (*Figure 7G,H*). This suggests that the Chinmo/Imp module is necessary to maintain the unlimited mitotic potential of NBs stalled in an early temporal identity (*Figure 7H*). Together with our results showing that over-expression of Chinmo is sufficient to induce persistence of NBs in adults (*Figure 3H*), these experiments also indicate that silencing of *Imp* and *chinmo* by the temporal patterning system may be necessary to limit

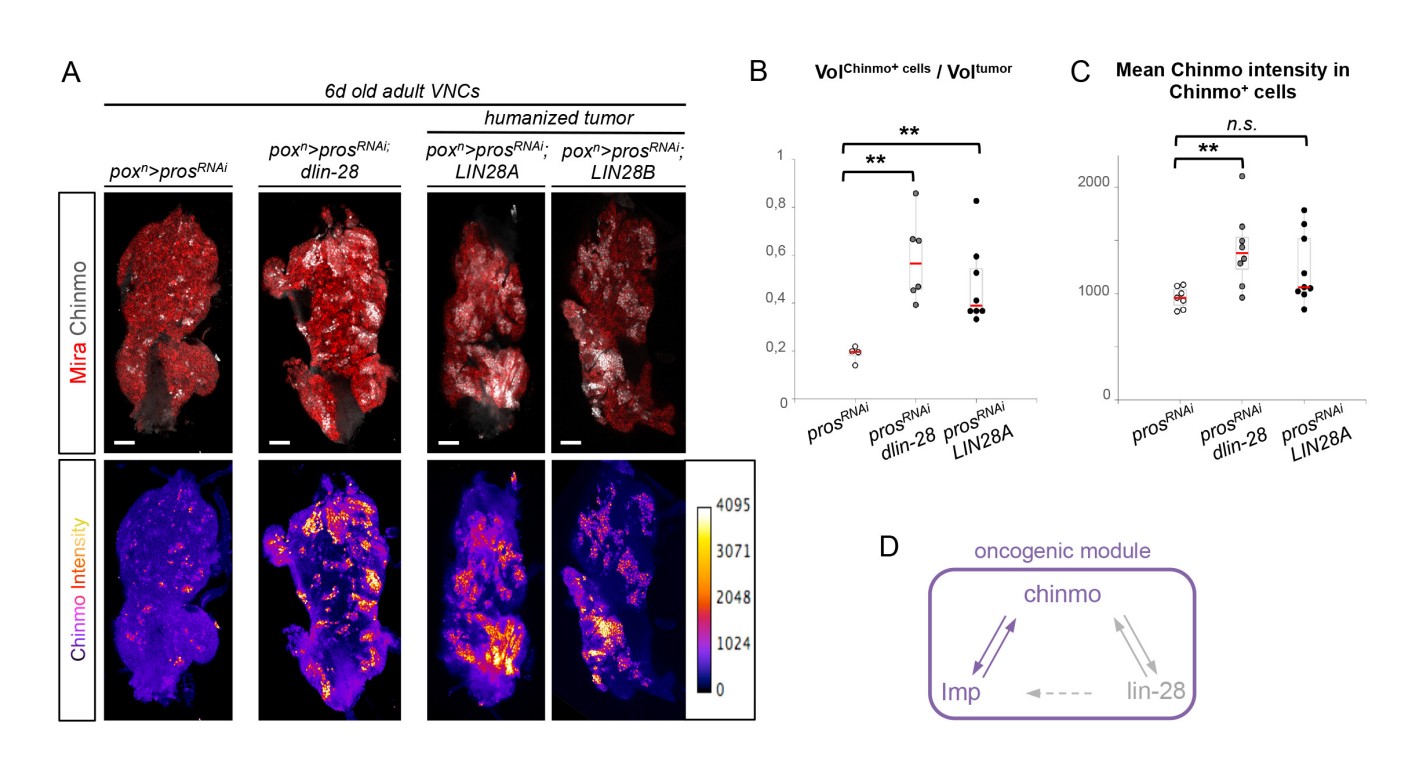

**Figure 6.** Chinmo, Imp and Lin-28 form an oncogenic loop. (A) Chinmo expression in 5 day-old adult VNCs color-coded relative to staining intensity. All transgenes are expressed with the $pox^n$-GAL4 driver. Humanized tumors mis-express human LIN28A or LIN28B. (B) Ratio representing the volume of Chinmo$^+$ cells over the total tumor volume in 5 day-old adult VNCs. $pox^n$>$pros^{RNAi}$ (n = 4 VNCs, m = 0.188, SEM = 0.017), $pox^n$>$pros^{RNAi}$, dlin-28 (n = 6 VNCs, m = 0.583, SEM = 0.072) and $pox^n$>$pros^{RNAi}$, LIN28A (n = 8 VNCs, m = 0.474, SEM = 0.060). p-values are respectively 9.5x10$^{-3}$ and 8.0x10$^{-3}$. (C) Mean Chinmo intensity in Chinmo$^+$ cells. $pox^n$>$pros^{RNAi}$ (n = 7 tumors, m = 962, SEM = 37), $pox^n$>$pros^{RNAi}$, dlin-28 (n = 8 tumors, m = 1413, SEM = 125) and $pox^n$>$pros^{RNAi}$, UAS-LIN28A (n = 9 tumors, m = 1236, SEM = 110). Each sample is the mean of 3 different focal sections of the same tumor. p-values are respectively 5.9x10$^{-3}$ and 7.1x10$^{-2}$. (D) Representation of the observed cross-regulatory interactions composing the oncogenic module.

The following figure supplements are available for figure 6:

**Figure supplement 1.** Lin-28 is dispensable for tumor growth.

**Figure supplement 2.** Lin-28 mis-expression is not sufficient to initiate tumors.

**Figure supplement 3.** Lin-28 positively regulates *chinmo* and *Imp* in tumors. Overexpression of *lin-28* in $pox^n$>$pros^{RNAi}$ tumors leads in adults to an increase in the number of Chinmo$^+$ cells, all of which also express Imp.

**Figure supplement 4.** Schematic conclusions.

NB mitotic activity and schedule the competency for NB terminal differentiation during metamorphosis (**Figure 7H**).

Surprisingly, in contrast to tumors, Chinmo knock-down in $svp^{-/-}$ MARCM clones does not lead to the down-regulation of Imp (**Figure 7G** and **Figure 7—figure supplement 2B**). This shows that *Imp* expression in early-identity NBs does not require Chinmo. Thus, expression of *chinmo* and *Imp* in NBs during development and tumorigenesis are not subordinated to the same cross-regulatory interactions.

## Malignant tumors can only be induced during the early CIL$^+$ expression window

We then tested whether ectopic expression of CIL proteins in $pros^{-/-}$ tumors was due to aberrant maintenance from an early CIL$^+$ cell of origin. We induced $pros^{-/-}$ tumors by MARCM in the VNC

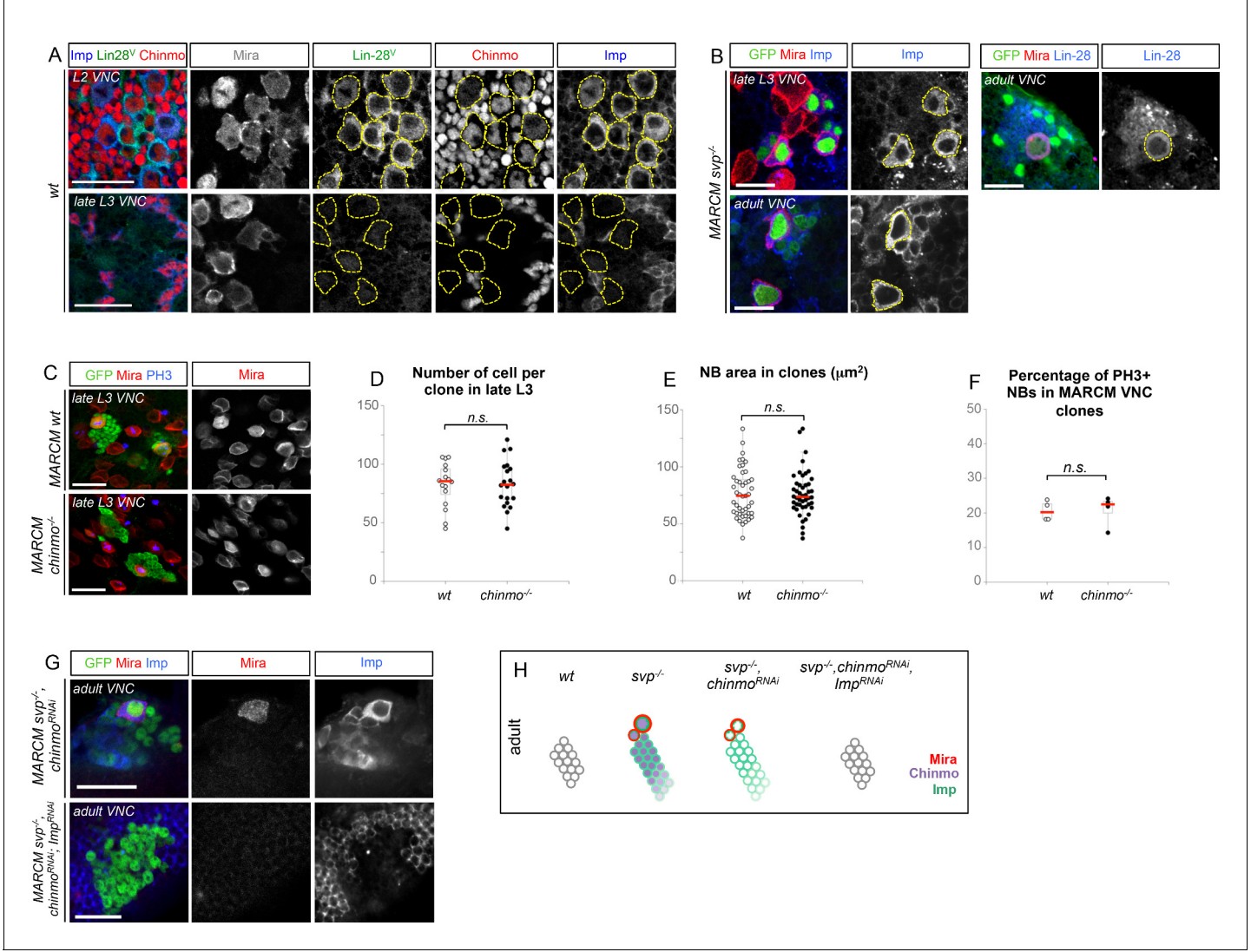

**Figure 7.** The temporal series silences *chinmo*, *Imp* and *lin-28* in NBs for their timely termination during development. (**A**) Chinmo, Lin-28 and Imp are coexpressed in *wt* VNC NBs in L2, and are silenced in NBs in late L3. Note that in late L3, surrounding early-born neurons keep expressing Imp and Chinmo whereas Lin-28 is down-regulated in all cells. (**B**) Imp is maintained in L1-induced MARCM *svp^-/-* NBs (GFP+) in late L3. Lin-28 and Imp are maintained in MARCM *svp^-/-* NBs (GFP+) that persist in adult. (**C**) *wt* and *chinmo^-/-* MARCM clones induced during embryogenesis. (**D**) Number of cells per clone in late L3 in VNC *wt* and *chinmo^-/-* MARCM clones induced during embryogenesis. *wt* MARCM 40A (n = 16 clones, m = 82, SEM = 4.7); *chinmo^-/-* MARCM clones (n=17 clones, m = 83, SEM = 4.8). p-values is 0.88. (**E**) Mean NB area in late L3 VNC *wt* and *chinmo^-/-* MARCM clones induced during embryogenesis. *wt* MARCM 40A (n = 45 NBs, m = 76.7, SEM = 3.2), *chinmo^-/-* MARCM clones (n = 46 NBs, m = 76.3, SEM = 2.9). p-value is 0.88. (**F**) Mean percentage of PH3+ NBs in late L3 VNC MARCM *wt* and *chinmo^-/-* clones induced during embryogenesis. *wt* clones (n = 4 VNCs, m = 20.6, SEM = 1.43), *chinmo^-/-* clones (n = 4 VNCs, m = 20.8, SEM = 2.23), p-value is 0,88. (**G**) NBs persist in adult MARCM *svp^-/-*, *chinmo^RNAi* clones induced in L1. NBs are smaller (***Figure 7—figure supplement 1A***) and maintain Imp expression. Removing both Chinmo and Imp (*chinmo^RNAi*, *Imp^RNAi*) in a *svp^-/-* MARCM clone is sufficient to restore NB elimination before the end of development. (**H**) Schematic recapitulation of the above experiments.

The following figure supplements are available for figure 7:

**Figure supplement 1.** Lin-28 and Chinmo are respectively silenced in the CNS prior to and during metamorphosis, whereas Imp remains expressed in a subset of adult neurons.

**Figure supplement 2.** Chinmo promotes the long-term growth of NBs stalled in an early temporal identity but is not required for Imp expression.

either early (during L1/L2) when Chinmo is still expressed in NBs and their progeny, or at midL3 after the CIL module has been switched off by temporal factor progression. One day after induction, ectopic Chinmo is observed in 91% of L1/L2-induced tumors (n = 65). In contrast, Chinmo is not detected 1 day after induction in midL3-induced tumors (n = 54) (*Figure 8A*), or 3 days after induction when larvae are grown in the sterol-free medium that prevent pupariation (26 out of 27 tumors still lack Chinmo) (*Figure 8B*, *Figure 8—figure supplement 1*). Thus, Chinmo$^+$ dNBs are only present in *pros*$^{-/-}$ tumors initiated during an early window of development (before midL3).

We then investigated the malignant potential of L1/L2- and midL3-induced tumors. L1/L2-induced *pros*$^{-/-}$ clones persist in 100% of adults and rapidly progress to cover the totality of the VNCs (n>100). In contrast, midL3-induced tumors persist in only 36.4% of adult VNCs (n = 19) (*Figure 8C*), and make an average of 0.63 persisting tumor per adult VNC (SEM = 0.23), deriving from an average of 13.5 tumors per larval VNCs (n= 16, SEM = 1). These rare persisting tumors (4,7%) in adult VNCs remain small, containing no more than 100 dNBs. Thus, most midL3-induced tumors differentiated during metamorphosis, and the few tumors that persist in adults possess a limited growth potential (*Figure 8C*). This limited growth potential of midL3-induced tumors was also observed when larvae were maintained in the sterol-free diet for 7 days after induction. In such conditions, midL3-induced tumors rapidly stop growing, with many dNBs progressively losing GFP expression, suggesting reduced transcriptional activity, and exhibiting reduced mitotic index and cytoplasmic extensions typical of quiescent NBs indicating a progressive exhaustion of their mitotic potential (*Chell and Brand, 2010*; *Sousa-Nunes et al., 2011*; *Truman and Bate, 1988*) (*Figure 8D*, *Figure 8—figure supplement 1, 2A,B*). Similar features are observed in surrounding *wt* NBs that do not undergo terminal differentiation in the absence of pupal signals (*Figure 8—figure supplement 1*, and *Figure 8—figure supplement 2B,C*). In contrast, L1/L2-induced tumors remain highly proliferative leading to massive dNB tumors covering the whole CNS (*Figure 8D*). Together these results indicate that malignant *pros*$^{-/-}$ tumors can only be induced if they are initiated during an early window of development (before midL3) while NBs and their progeny express the CIL module. In contrast, dNBs induced by the dedifferentiation of late-born CIL-negative GMCs cannot reactivate Chinmo expression, possess a limited proliferation potential and undergo terminal differentiation during metamorphosis like *wt* NBs, or persist as benign, slow-growing, tumors in adults.

## The temporal identity of a cell determines its malignant susceptibility

These results suggest that the silencing of the CIL genes in late-born neural cells by temporal patterning progression may abolish their malignant potential upon dedifferentiation. Alternatively, absence of CIL reactivation in late-induced tumors could be due to a non-permissive microenvironment subsequent to developmental progression. To distinguish these possibilities, we combined the GAL80$^{ts}$ system (*McGuire et al., 2004*) and MARCM to temporally control the induction of *pros*$^{RNAi}$ tumors in *svp*$^{-/-}$ NB clones blocked in an early temporal identity. Loss of Svp was induced in L1 by a 45 min heat-shock. Then larvae were kept at 18°C in the control experiment, or switched to 29°C from late L3 or pupal stages for late induction of *pros*$^{RNAi}$. While adult flies raised at 18°C possess a single NB per clone due to the early loss of Svp, flies raised at 29°C from L3 or pupal stages contain large tumors of CIL$^+$ dNBs in their VNCs (*Figure 9A,B*). Thus, the GMCs of NBs blocked in an early temporal identity remain predisposed to generate CIL$^+$ malignant tumors independently of the developmental stage. These results demonstrate that the malignant potential of a neural cell undergoing dedifferentiation is dictated by its temporal identity, and blocking temporal factor progression in NBs extends the window of malignant susceptibility.

## Discussion

In this study, we have uncovered that the NB-encoded temporal patterning system delineates an early window of malignant susceptibility (*Figure 9B*) through the regulation of an early growth module involving Chinmo, Imp/IGF2BP and Lin-28. Here, we discuss how our work supports an ancestral model that may explain the rapid malignant progression of neural tumors induced during early development.

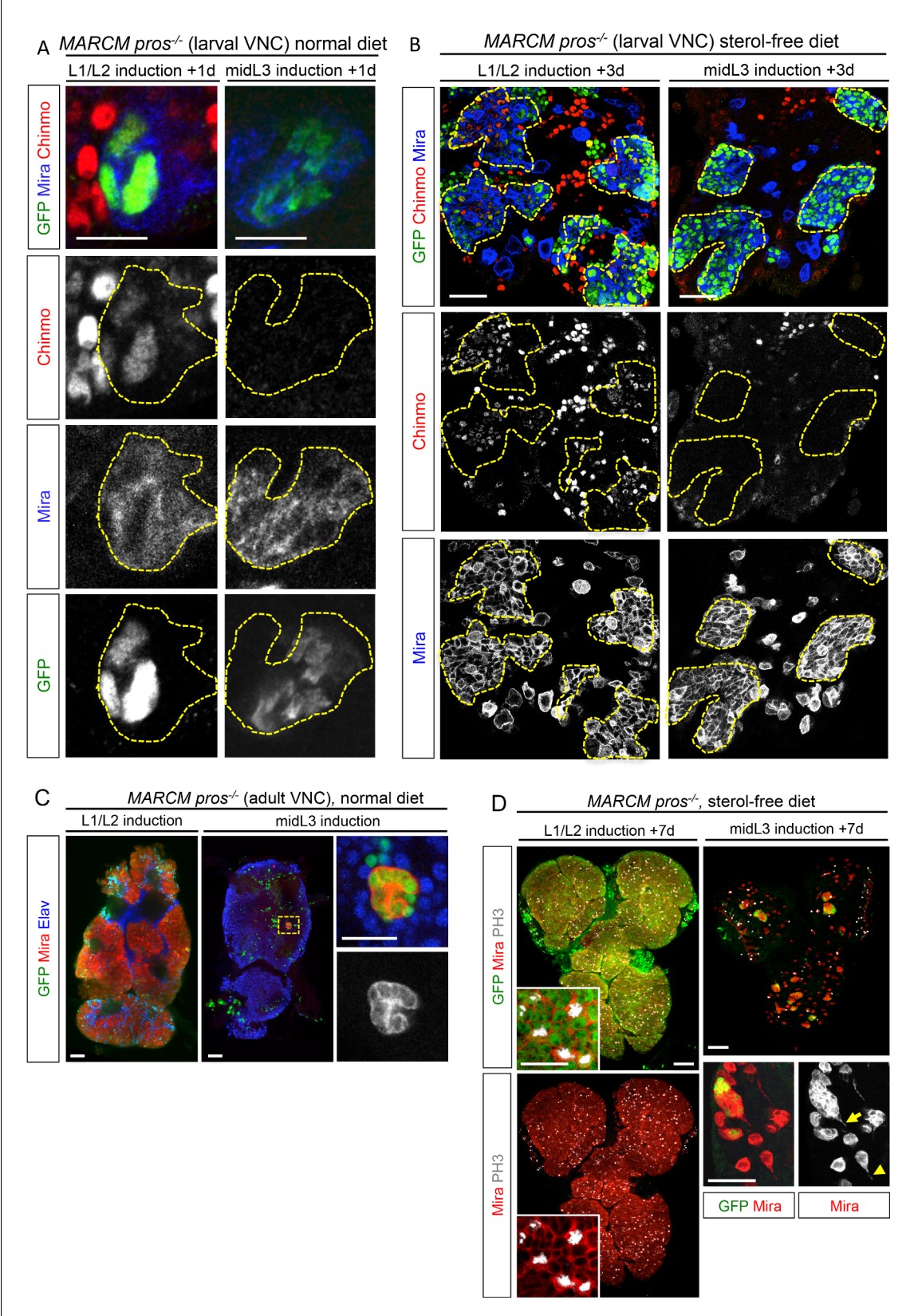

**Figure 8.** *pros*[-/-] tumors induced in the VNC of midL3 larvae do not express Chinmo and do not grow in adults. (**A**) One day after early (L1/L2), induction most dNBs from *pros*[-/-] clones retain Chinmo expression. In contrast, Chinmo is absent from dNBs 1 day after late (midL3) clonal induction. *Figure 8 continued on next page*

*Figure 8 continued*

(**B**) Three days after L1/L2-induction in larvae raised on the sterol-free diet, *pros*-/- clones contain Chinmo+ dNBs (54 out of 59). In contrast, 3 days after mid-L3 induction on the sterol-free diet, *pros*-/- clones do not contain Chinmo+ dNBs (26 out of 27). (**C**) Adult VNC containing L1/L2-induced *pros*-/- MARCM clones are covered by tumors. In contrast, midL3-induced *pros*-/- MARCM clones are rare and remain small in adult VNCs. (**D**) On the sterol-free diet, 7 days after L1/L2-induction, *pros*-/- clones keep proliferating and generate large tumors of GFP+ dNBs (Mira shown in red) that cover the whole CNS. In contrast, *pros*-/- clones, 7 days after midL3-induction, rapidly stop growing and dNBs exhibit quiescence markers such as the loss of GFP and cytoplasmic extensions (arrow emphasizes cytoplasmic extensions from dNBs, arrowhead emphasizes a cytoplasmic extension from a *wt* NB).

The following figure supplements are available for figure 8:

**Figure supplement 1.** Schematic conclusion. Schematic representation recapitulating the conclusions from experiments in *Figure 8*.

**Figure supplement 2.** The proliferation potential of normal NBs decreases over time while dNBs continue proliferating.

## An oncogenic module defining growth-sustaining cells in neural tumors with early developmental origins

We have demonstrated that the unlimited growth potential of *pros*-/- and *brat*-/- tumors induced during early larval development relies on the aberrant maintenance of an oncogenic module respectively co-opted from dedifferentiating early-born GMCs and INPs (and possibly neurons in *nerfin1*-/- tumors). The core components of the module involve the BTB transcription factor Chinmo and the mRNA-binding protein Imp. Chinmo and Imp positively cross-regulate at the transcriptional and translational level respectively. We find that Chinmo is a proto-oncogene that is able to promote the transcription of a large set of genes that boost protein synthesis and cell-cycle progression, and is also a strong repressor of neural differentiation. Consequently, over-expression of Chinmo in NBs and GMCs, but not in neurons (*Zhu et al., 2006*), is sufficient to cause NB amplification and tumor growth. This is consistent with a previously identified oncogenic activity of Chinmo when expressed in the hematopoietic and imaginal disc precursors (*Doggett et al., 2015*; *Flaherty et al., 2010*). In contrast, Imp mis-expression is not sufficient to trigger NB amplification but Imp is necessary to maintain *pros*-/- tumor growth beyond developmental stages, at least partly by allowing/sustaining Chinmo expression. Another mRNA-binding protein, Lin-28, is also positively regulated by Chinmo in tumors. We found that Lin-28 is however dispensable for tumor growth although Lin-28 over-expression enhances the proportion of Chinmo+ dNBs in tumors. Thus Lin-28 overexpression may promote the self-renewal of Chinmo+, Imp+ dNBs and therefore increase the growth potential of the tumor. This role is consistent with the observation that high expression of LIN28 isoforms in human tumors promotes tumor growth (*Viswanathan et al., 2009*). Interestingly, a recent RIP-seq analysis in *Drosophila* embryos has uncovered both Imp and chinmo mRNAs among the most highly enriched targets of Lin-28 (*Chen et al., 2015a*), and mammalian Imp and Lin-28 orthologs share many common RNA targets (*Yang et al., 2015*). Thus, Imp and Lin-28 may also directly co-regulate Chinmo translation in tumors. Further investigation is required to identify all components and the regulatory principles of this oncogenic network.

Collectively, our results imply that, although representing a minor sub-population of dNBs, the subset of dNBs co-expressing Chinmo, Imp, and Lin-28 (CIL+ dNBs) is sustaining the unlimited growth of the tumor. It remains to be demonstrated whether these CIL+ dNBs act as cancer stem cells (CSCs) (*Beck and Blanpain, 2013*), able to self-renew and generate the bulk of the tumor, or represents a transient amplifying population of progenitors born from another population of rare and slow-proliferating CSCs.

## Temporal regulation of NSC mitotic potential during development

We have shown that Chinmo, Imp and Lin-28 are co-expressed in early NBs and their progeny and are silenced shortly after the L2/L3 transition in ageing NBs by progression of the temporal transcription factor series. Surprisingly, although Chinmo promotes tumor growth during both larval and adult stages, loss of Chinmo in normal NBs did not significantly affect the total number of progeny generated at the end larval stages. This indicates that the growth and proliferation of normal NBs and dNBs does not exhibit the same dependency on Chinmo.

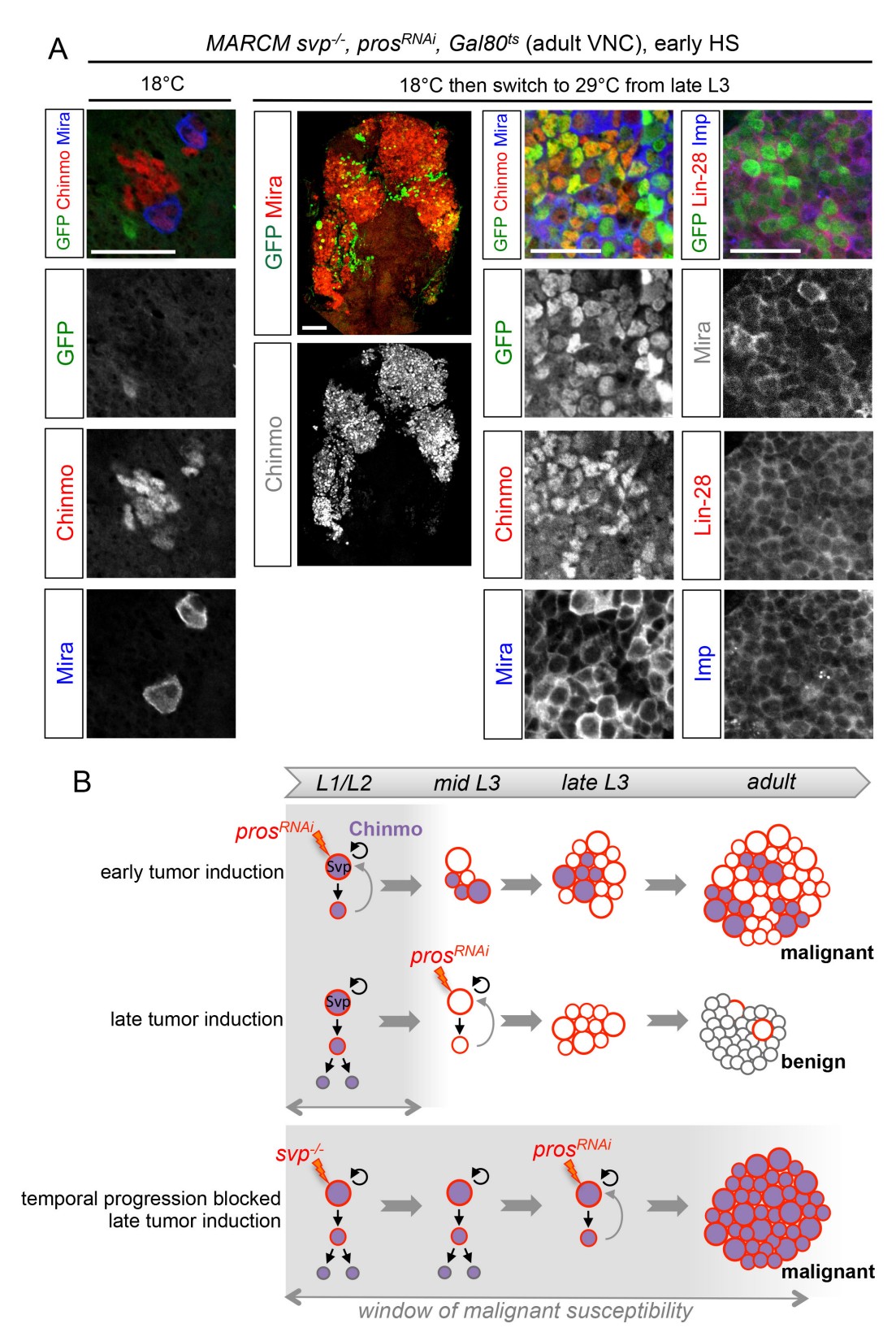

**Figure 9.** The temporal series regulates the malignant susceptibility of neural cells born during development. (**A**) Larvae were raised at 18°C and heat-shocked in L1 to induce *svp*$^{-/-}$ MARCM clones. Controls were kept at 18°C, to prevent *pros*$^{RNAi}$ expression, leading to the persistence of a single NB

*Figure 9 continued on next page*

*Figure 9 continued*

per clone in adults. In contrast, if larvae are switched to 29°C from late L3, *pros^RNAi* is expressed leading to large tumors in adults, exclusively composed of dNBs expressing Chinmo, Imp and Lin-28. (**B**) Schematic recapitulation of the above experiments. Blocking temporal progression in NBs extends the window of malignant susceptibility.

Although Chinmo is not required for proper NB mitotic activity in larvae, its over-expression in NBs is sufficient to promote tumorigenic growth in adults. Moreover, NBs blocked in an early temporal identity fail to persist in adults if lacking Imp and Chinmo. Thus, our results suggests that the silencing of Chinmo and Imp in early L3 by the temporal series is a way to limit the mitotic potential of NBs and ensure their timely termination during pupal stages. The role of Lin-28 in regulating NB mitotic activity during development is still unknown, but in mice, Imp1/Igf2bp1 and Lin28a/b are co-expressed in fetal cortical progenitors where they post-transcriptionally regulate growth genes involved in the PI3K/TOR pathway (*Yang et al., 2015*). This suggests an evolutionary conserved growth-promoting role of these factors during neural development. Moreover, Lin28a and Imp1 are downregulated in mammals before birth (*Yang et al., 2015*). This temporal pattern of expression is reminiscent to what is observed in NBs during *Drosophila* larval stages and suggests that the growth dynamic of fetal NSCs in mammals and larval NBs are controlled by evolutionary conserved gene networks. However, whether an analogous temporal transcription factor series in mammals similarly regulates Imp1 and Lin28a/b remains to be demonstrated.

We noticed that loss of Chinmo in temporally blocked NBs does not induce the silencing of Imp. In contrast, Chinmo is necessary for Imp maintenance in tumors. Thus, *chinmo* and *Imp* are not subordinated to the same regulatory mechanisms whether in a developmental or in a tumorigenic context. This indicates that regulatory mechanisms upstream of *chinmo* and *Imp* may be different in normal NBs (the temporal series) and in dNBs after co-option.

## A NSC-encoded clock that regulates malignant susceptibility during early development

We have shown that progression throughout the NB-encoded series of temporal transcription factors terminates an early window of malignant susceptibility in the developing CNS around the beginning of the L3 stage. GMCs born during this early developmental window can generate malignant tumors upon loss of Pros. In contrast, GMCs born after the early window generate benign tumors that either completely differentiate during metamorphosis, demonstrating sensitivity to the growth-terminating cues operating in pupae, or persist in adults but fail to consistently grow, demonstrating a limited growth potential. Blocking progression of the temporal patterning system from early larval stages extends the window of susceptibility and late-born GMCs remain prone to malignant transformation, up to (and possibly beyond) metamorphosis. Our clonal analysis indicates that this window of malignant susceptibility correspond to the temporal window of expression of Chinmo and Imp. All together, our data strongly suggest that it is the presence of an activated Chinmo/Imp module in the cell (GMC) of origin that confers malignant potential. Thus, the temporal patterning regulates malignant susceptibility, at least partly, through the temporal control of Chinmo and Imp in newly-born GMCs during neurogenesis. Whether embryonic GMCs are prone to malignant transformation, like early larval GMCs, remains unclear. We could not in our various assays limit the loss of Pros to embryonically-born GMCs. This question therefore requires further investigation.

Our work reveals a model for the rapid malignant progression of tumors with an early developmental origin. First, inactivation of genes that govern the terminal differentiation of intermediate progenitors or immature neurons triggers exponential NSC amplification. Second, if occurring at an early developmental stage, this dedifferentiation process interferes with the NSC-encoded temporal program that normally limits mitotic potential by silencing genes like Chinmo, Imp and Lin-28 in *Drosophila*. As a result, Chinmo, Imp and Lin-28 (or their analogs in humans) are co-opted in dedifferentiated cells to unleash an early mode of growth that is resistant to differentiation cues during late development and sufficient to sustain unlimited NSC amplification and malignant progression. In contrast to tumors initiated during adulthood, the cells of origin (CIL[+] GMCs in the case of *pros^-/-* tumors) already express the early oncogenic modules that are sufficient to sustain rapid growth. Consequently, early-induced tumors do not need to accumulate novel mutations to progress to

malignancy. In contrast, later-born neural cells that have already silenced proto-oncogenic, modules may require the accumulation of more genetic and epigenetic alterations for the reestablishment of oncogenic combinations causing malignancy. All together, our data uncovers the developmental program regulating the malignant susceptibility of neural cells in *Drosophila*, and provides a model that may help to unveil the molecular basis underlying the rapid malignant growth of neural tumors with early developmental origins.

## Deciphering temporal patterning in neural progenitors to understand human tumors with early developmental origins

A recent study in mice has shown that inactivation of *Smarcb1* can most efficiently cause tumors resembling atypical teratoid/rhabdoid tumours (AT/RTs), an agressive pediatric CNS cancer, when induced during an early window of pre-natal development (E6 to E10) (*Han et al., 2016*). In human, the embryonic/fetal origin of tumor cells is also suspected for a number of other pediatric neural cancers such as medulloblastoma in the cerebellum, retinoblastoma in the eye, and neuroblastoma in the sympathetic nervous system (*Marshall et al., 2014*). Such tumors are typically composed of cells with immature characteristics, referred to as embryonal (*Marshall et al., 2014*), and they usually carry few genetic alterations (*Vogelstein et al., 2013*). However, the mechanisms underlying their rapid malignant transformation remains largely unsolved.

We have demonstrated that the ability of human LIN28A/B to promote Chinmo expression in tumors is conserved in *Drosophila*, indicating that some regulatory aspects of the Chinmo/Imp/Lin-28 oncogenic module are likely to be evolutionary conserved. Interestingly, a recent survey of the scientific literature has revealed that *LIN28A/B* genes are much more frequently expressed in pediatric cancers than adult ones (*Carmel-Gross et al., 2016*), a feature often associated with poor prognosis (*Molenaar et al., 2012*; *Zhou et al., 2013*). Human IMPs/IGF2BPs proteins have been less extensively investigated but are expressed in neuroblastoma and other cancers and are also associated with a poor outcome (*Bell et al., 2013*; *2015*). MYCN is a transcription factor of the MYC family, that like Chinmo in *Drosophila*, is expressed during early neurogenesis and promotes ribosome biogenesis and protein synthesis (*Boon et al., 2001*; *Knoepfler et al., 2002*; *Wiese et al., 2013*). Interestingly, it is known to be up-regulated in many pediatric cancers of neural origin (*Huang and Weiss, 2013*; *Swartling et al., 2010*; *Theriault et al., 2014*), and is positively regulated by LIN28 and IMPs/IGF2BPs in neuroblastoma (*Bell et al., 2015*; *Cotterman and Knoepfler, 2009*; *Molenaar et al., 2012*). Thus, Chinmo/Imp/Lin-28 in insect may compose an ancestral oncogenic module with similar functions and regulatory interactions as mammalian MYCN/IMP/LIN28 during early neural development and tumorigenesis.

Whether the temporal expression pattern of such modules, or oncofetal genes in general, delineates windows of malignant susceptibility in mammals is not clear. Interestingly, retinoblastoma is caused by the loss of the Retinoblastoma (Rb) protein that triggers dedifferentiation of photoreceptor cone cells (*Xu et al., 2014*). Cone cells are among the earliest progeny to be generated by retinal progenitors (from E12 to E16 in mice) and their birth-order is dictated by a series of sequentially expressed temporal transcription factors (*Mattar et al., 2015*; *Young, 1985*). In human, maturing cone cells in the retina, but not later born photoreceptors, express high levels of MYCN. Moreover, cooption of MYCN appears instrumental for retinoblastoma tumor growth (*Xu et al., 2009*). Therefore, temporal patterning in retinal progenitors, as in *Drosophila* NBs, may dictate the malignant susceptibility of their progeny according to their birth order through the regulation of an early growth module involving MYCN. Our work suggests that deciphering temporal specification mechanisms in the different regions of the nervous system will help identify the cell types and gene networks at the origin of pediatric neural cancers.

## Materials and methods

### Fly culture

*Drosophila* stocks were maintained at 18°C on standard medium (8% cornmeal/8% yeast/1% agar). Sterol-free fly food was obtained by replacing classical yeast strain by a sterol-mutant *erg2* knock-out strain in the medium (*Katsuyama and Paro, 2013*; *Parkin and Burnet, 1986*).

## Image processing

Confocal images were acquired on Zeiss LSM510 and Zeiss LSM780. ImageJ, FIJI and Photoshop were used to process confocal data. The area of individual tumors was measured from a z-projection of the entire tissue.

## Statistical analysis

For each experiment, at least 3 biological replicates were peformed.

Biological replicates are defined as replicates of the same experiment with flies being generated by different parent flies.

For all experiments, we performed a Mann-Whitney test for statistical analysis, except for *Figure 7D* and *Figure 3—figure supplement 2* where a Fisher's exact test was used.

No data were excluded.

Statistical test were performed with the online BiostaTGV (http://marne.u707.jussieu.fr/biostatgv/).

Results are presented as dot plots, also depicting the median in red and a boxplot in the background (Whisker mode : 1.5IQR).

The sample size (n), the mean (m), the standard error of the mean (SEM), and the p-value are reported in the Figure legends.

****: p-value $\leq$ 0.0001, ***: p-value $\leq$ 0.001, **: p-value $\leq$ 0.01 and *: p-value $\leq$ 0.05.

## Fly lines

Experiments were performed at various temperatures as stated below. For generating MARCM clones (*Lee and Luo, 1999*), the following driver stocks were used:

- *FRT19A, tubP-GAL80, hs-FLP1 w; tub-GAL4, UAS-mCD8-GFP/CyO$_{Act}$GFP*
- *elav-GAL4, UAS-mCD8-GFP, hs-FLP$^{122}$; FRT40A, tubP-GAL80/CyO*
- *hs-FLP$^{122}$, tubGal4, UAS-GFP; tub-GAL80, FRT40A; Δlin-28*
- *hs-FLP$^{122}$, tubGal4, UAS-GFP; tub-GAL80 FRT80B*
- *w, tub-GAL4, UAS-nlsGFP::6xmyc::NLS, hsFLP$^{122}$; FRT82B, tubP-GAL80/TM6B*

They were crossed with the following stocks:

- *FRT19A, Imp$^7$/FM6* (*Munro et al., 2006*) (from F. Besse)
- *chinmo$^1$, UAS-mCD8-GFP, FRT40A/CyO* (*Zhu et al., 2006*)
- *brat$^{11}$, FRT40A/CyO$_{Act}$GFP* (from B. Bello)
- *FRT2A, Df(3L)nerfin-1$^{159}$/TM6b* (from L. Cheng (*Froldi et al., 2015*))
- *FRT82B, pros$^{17}$/TM6* (from Bloomington #5458)
- *FRT82B, cas$^{24}$/TM6* (*Maurange et al., 2008*)
- *FRT82B, pros$^{17}$, cas$^{24}$/TM6*
- *FRT82B, svp$^{e22}$/TM6* (from Bloomington #6190).
- *Δlin-28, FRT80B/TM6* (*Chen et al., 2015a*)
- *UAS-pros$^{RNAi}$/CyO$_{Act}$GFP; Δlin-28, FRT80B/TM6*
- *brat$^{11}$, FRT40A/CyO$_{Act}$GFP; Dlin-28/TM6*

The progeny of the above crosses were heat-shocked 1 hr at 37°C just after larval hatching and raised at 25°C.

- *FRT82B, svp$^{e22}$, UAS-chinmo$^{RNAi1}$/TM6* (*UAS-chinmo$^{RNAi1}$* from TRiP #HMS00036, Bloomington #33638)
- *UAS-ImpRNAi1; FRT82B svpe22, UAS-chinmoRNAi1/TM6* (*UAS-ImpRNAi1*from VDRC #20322).

The progeny of the above crosses were heat-shocked 1 hr at 37°C just after larval hatching and raised at 29°C.

- *UAS-pros$^{RNAi}$; FRT82B, svp$^{e22}$, Gal80ts/TM6* (*UAS-pros$^{RNAi}$* from VDRC #101477)

The progeny of this cross were raised at 18°C, heat-shocked 1 hr at 37°C just after larval hatching, and either maintained at kept at 18°C (restrictive temperature) for the rest of development (controls) or switched at 29°C (permissive temperature) in late L3 stage or early pupae (*Figure 9A*).

Flip-out clones were generated using *hs-FLP; Act5c>CD2>GAL4, UAS-GFP* (from N. Tapon) with:

- *UAS-chinmo* (Bloomington #50740)

The progeny of this cross were heat-shocked 1 hr at 37°C just after larval hatching and raised at 25°C.

- *UAS-pros$^{RNAi}$ UAS-chinmo$^{RNAi1}$*(TRiP#HMS00036, Bloomington #33638)

The progeny of this cross were heat-shocked 1 hr at 37°C during L2 stages and raised at 29°C. The GAL4 lines used were the following:

- *nab-GAL4* (#6190 from Kyoto DGRC) is a GAL4 trap inserted into *nab* (CG33545) that is active in all NBs of the VNC and central brain from late embryogenesis (*Maurange et al., 2008*)
- *eagle-GAL4* (*eg-GAL4*, Bloomington #8758) is active in a small subset of NBs in the VNC and brain.
- *pox$^n$-GAL4* (*Boll and Noll, 2002*) is active in 6 NBs of the VNC.

The UAS lines used were:

- *UAS-chinmo* (Bloomington #50740)
- *UAST-GFP-Imp* (*Medioni et al., 2014*)
- *UAS-pros$^{RNAi}$* (TRiP#JF02308,Bloomington #26745)
- *UAS-pros$^{RNAi}$* (VDRC #101477)
- *UAS-Imp$^{RNAi1}$*(VDRC #20322)
- *UAS-Imp$^{RNAi2}$* (VDRC #20321)
- *UAS-chinmo$^{RNAi1}$*(TRiP#HMS00036, Bloomington #33638)
- *UAS-chinmo$^{RNAi2}$*(NIG-Fly #17156R-1)
- *UAS-lin-28/CyO* (*Chen et al., 2015a*)
- *UAS-LIN28A* (*Chen et al., 2015a*)
- *UAS-LIN28B* (*Chen et al., 2015a*)
- *UAS-dicer2* (Bloomington #24650 and #24651) was used in combination with GAL4 lines in order to improve RNAi efficiency.

The *lin28$^{\Delta 1}$, {lin-28::Venus}* (Lin-28-V) stock contains a genomic rescue transgene encoding a fluorescently tagged Lin-28 (*Chen et al., 2015a*).

GFP-Imp is the protein-trap line #G0080 (*Morin et al., 2001*).

The progeny of the above crosses were raised at 29°C.

*pox$^n$-GAL4* is already active in embryonic NBs (*Boll and Noll, 2002*). However, we could show that inhibition of embryonic GAL4, using a tub-*Gal80$^{ts}$* transgene, does not significantly alter tumor formation demonstrating that *pros$^{RNAi}$* expression during early larval stages is sufficient to cause malignant tumors in adults.

## Immunohistochemistry

Dissected tissues were fixed 5 min or more in 4% formaldehyde/PBS depending on the primary antibody. Stainings were performed in 0.5% triton/PBS with antibody incubations separated by several washes. Tissues were then transferred in Vectashield (Cliniosciences, France) with or without DAPI for image acquisition. Primary antibodies were: chicken anti-GFP (1:1000, Tebubio, France), mouse anti-Mira (1:50, A. Gould), guinea-pig anti-Mira (1:1000, A. Wodarz), guinea-pig anti-Asense (1:1000, J. Knoblich), rabbit anti-PH3 (1:500, Millipore, Billerica, MA), rat anti-PH3 (1:500, Abcam, UK), rat anti-Elav (1:50, DSHB, Iowa City, IA), rat anti-Chinmo (1:500, N. Sokol), rabbit anti-Castor (1:500, W. Odenwald), guinea-pig anti-Hunchback (1:500, J. Reinitz), guinea-pig anti-Kruppel (1:500, J. Reinitz), rabbit anti-Pdm (1:500, X. Yang), mouse anti-Svp (1:50, DSHB), rabbit anti-Imp (1:500, P. Macdonald), rat anti-Lin-28 (1:500, N. Sokol). Adequate combinations of secondary antibodies (Jackson ImmunoResearch, West Grove, PA) were used to reveal expression patterns.

## RNA extraction

Females from the driver line *nab-Gal4, UAS-dicer2* were crossed to males carrying *UAS-pros$^{RNAiVDRC}$ 2) UAS-pros$^{RNAiVDRC}$; UAS-chinmo$^{RNAiTRiP}$*and *UAS-pros$^{RNAiVDRC}$; UAS-chinmo*. Crosses were grown 7 days at 18°C and then switched at 29°C. Late L3 VNCs were dissected in cold PBS 3 days after the 29°C switch, during 30 min dissection rounds. Dissected VNCs were put in 500μL cold Lysis Buffer (RA1 from the Total RNA Isolation kit, Macherey Nagel, Germany) supplemented with 50 μL glass beads (diameter 0.75-1mm, Roth, A554.1) and frozen in liquid nitrogen at the end of the dissection

round. Sample tubes were then stored at -80°C up to RNA extraction. Biological triplicates were made for each condition with brain numbers as follows (1) n=79, n=57, n=65; (2) n= 70, n=58, n=78; (3) n=51, n=60, n=82. For the RNA extraction, dissected brains stored in liquid nitrogen were thawed on ice. 10 µL TCEP were added to each tube following by 40s vortex. RNA extraction was then performed following the Total RNA Isolation NucleoSpin RNA XS protocol (Macherey Nagel). RNA quality and quantity were checked by running samples on an Experion RNA HighSens Chip (Biorad, 700–7105, Hercules, CA) and send to the Montpellier Genomix platform for RNA sequencing.

## Library preparation and sequencing

Libraries were constructed using the Truseq stranded mRNA sample prep kit (Illumina, ref.RS-122-2101, San Diego, CA) according to the manufacturer instructions. Briefly, poly-A RNAs were purified using oligo-d(T) magnetic beads from 300ng of total RNA. The poly-A$^+$ RNAs were fragmented and reverse transcribed using random hexamers, Super Script II (Life Technologies, ref. 18064–014, Waltham, MA) and Actinomycin D. During the second strand generation step, dUTP substitued dTTP. This prevents the second strand to be used as a matrix during the final PCR amplification. Double stranded cDNAs were adenylated at their 3' ends before ligation was performed using Illumia's indexed adapters. Ligated cDNAs were amplified following 15 cycles PCR and PCR products were purified using AMPure XP Beads (Beckman Coulter Genomics, ref.A63881, Brea, CA). Libraries were validated using a DNA1000 chip (Agilent, ref. 5067–1504, Santa Clara, CA) on a Agilent Bioanalyzer and quantified using the KAPA Library quantification kit (Clinisciences, ref. KK4824). Four libraries were pooled in equimolar amounts per lane and sequencing was performed on an HiSeq2000 using the single read protocol (50nt).

## RNA-Seq data analysis

Image analysis and base calling were performed using the HiSeq Control Software and Real-Time Analysis component provided by Illumina. The quality of the data was assessed using FastQC from the Babraham Institute and the Illumina software SAV (Sequence Analysis Viewer). Demultiplexing was performed using Illumina's sequencing analysis software (CASAVA 1.8.2). A splice junction mapper, TopHat 2.0.9 (*Kim et al., 2013*) (using Bowtie 2.1.0 [*Langmead and Salzberg, 2012*]), was used to align RNA-Seq reads to the Drosophila melanogaster genome (UCSC dm3) with a set of gene model annotations (genes.gtf downloaded from UCSC on March 6 2013). Final read alignments having more than 3 mismatches were discarded. Then, the gene counting was performed using HTSeq count 0.5.3p9 (union mode) (*Anders et al., 2014*). The data is from a strand-specific assay, the read has to be mapped to the opposite strand of the gene. Before statistical analysis, genes with less than 15 reads (cumulating all the analyzed samples) were filtered and thus removed. Differentially expressed genes were identified using the Bioconductor (*Gentleman et al., 2004*) package DESeq (*Anders and Huber, 2010*) 1.14.0. Data were normalized using the DESeq normalization method. Genes with adjusted p-value to less than 5% (according to the FDR method from Benjamini-Hochberg) were declared differentially expressed. To perform the functional analysis of the resulting list of genes with the Gene Ontology (GO) annotations, the topGO (*Alexa et al., 2006*) package from Bioconductor was used. Overrepresented GO terms were identified using Fisher's exact test with the weight method. As confidence threshold we used a p-value of 1%. To realize this analysis the differentially expressed genes were compared with those of all known genes present in the annotation. The GO categories were found in the Org.Dm.eg.db package (Carlson) based on the gene reporter EntrezGeneID. Gene Ontology and Kegg pathway analysis were performed using Flymine (*Lyne et al., 2007*). Raw RNA-seq data are available in the Gene Expression omnibus database (accession number: GSE64405).

## Transplantation experiments

Transplantations of *nab>GFP (UAS-GFP/+; nab-GAL4, UAS-dcr2/+), nab>pros$^{RNAi}$ (UAS-GFP/UAS-pros$^{RNAi}$; nab-GAL4, UAS-dcr2/+), nab>pros$^{RNAi}$; chinmo$^{RNAi}$ (UAS-GFP/UAS-pros$^{RNAi}$; nab-GAL4, UAS-dcr2/UAS-chinmo$^{RNAi}$)*, and *nab>chinmo (UAS-GFP; nab-GAL4, UAS-dcr2/UAS-chinmo)* VNCs and brains have been performed in *yw* females according to (*Rossi and Gonzalez, 2015*).

## Acknowledgements

We are grateful to EA Bach, B Bello, F Besse, L Cheng, C Doe, A Gould, J Knoblich, T Lee, P Macdonald, M Noll, W Odenwald, G Struhl, A Wodarz, X Yang and C Yu for flies and antibodies, and R Paro for the erg2 yeast strain. We also acknowledge the Bloomington Drosophila Stock Center (NIH P40OD018537), the Vienna Drosophila RNAi Center (VDRC), TRiP at Harvard Medical School (NIH/NIGMS R01-GM084947), Kyoto DGRC and NIG-Fly Stock Centers, and the Developmental Studies Hybridoma Bank (DSHB) for monoclonal antibodies. We thank France-BioImaging/PICsL infrastructure (ANR-10-INSB-04-01). We thank C Bertet, H Cremer, P Durbec and B Thompson for critical reading of the manuscript.

## Additional information

### Funding

| Funder | Grant reference number | Author |
|---|---|---|
| Centre National de la Recherche Scientifique | | Cédric Maurange |
| Fondation ARC pour la Recherche sur le Cancer | SFI20101201787 | Cédric Maurange |
| National Institutes of Health | R21OD019916 | Nicholas S Sokol |
| Fondation ARC pour la Recherche sur le Cancer | | Elodie Lanet |

The funders had no role in study design, data collection and interpretation, or the decision to submit the work for publication.

### Author contributions

KN-R, EL, CD, CM, Conception and design, Acquisition of data, Analysis and interpretation of data, Drafting or revising the article; SF, Acquisition of data, Analysis and interpretation of data; C-HC, Contributed unpublished essential data or reagents; HP, SR, Acquisition of data; NSS, Drafting or revising the article, Contributed unpublished essential data or reagents

### Author ORCIDs

Cédric Maurange, http://orcid.org/0000-0001-8931-1419

## Additional files

### Major datasets

The following dataset was generated:

| Author(s) | Year | Dataset title | Dataset URL | Database, license, and accessibility information |
|---|---|---|---|---|
| Lanet E, Foppolo S, Parrinello H, Rialle S, Maurange C, Narbonne-Reveau K, Dillard C | 2014 | The temporal patterning system governs the malignant susceptibility of neural progeny during Drosophila development | www.ncbi.nlm.nih.gov/geo/query/acc.cgi?acc=GSE64405 | Publicly available at NCBI Gene Expression Omnibus (accession no: GSE64405) |

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
