## [Decision Letter]

Thank you for submitting your work entitled "The temporal patterning system governs the malignant susceptibility of neural progeny during *Drosophila* development" for consideration by *eLife*. Your article has been reviewed by three peer reviewers, and the evaluation has been overseen by a Reviewing Editor (Hugo Bellen) and K VijayRaghavan as the Senior Editor.

The reviewers have discussed the reviews with one another and the Reviewing Editor has drafted this decision to help you prepare a revised submission.

Summary:

Early neural development is tightly controlled by a series of transcription factors that are expressed in a temporal fashion. In this study, the authors test the hypothesis that malignant transformation of neural progenitor cells occurs by disrupting the regulation of the temporal pattern of transcription factors that drive neuronal development. The authors find that early stage postembryonic neural progenitor cells, but not late stage neural progenitor cells, are prone to malignant transformation. Moreover, the authors present data that support a conserved role for Chinmo, Lin-28 and Imp for competency of progenitor cells to undergo malignant transformation. This is a novel and elegant study of broad interest, but there is room for improvement. The results are somewhat obscured by the fact that the paper is rather long and the genetics quite complex. The manuscript would benefit from shortening and bringing out the key results more clearly.

Main issues

The Abstract/Introduction/Discussion focuses on fetal origins of human pediatric neural tumors. However, the *Drosophila* data presented is mainly from postembryonic larval neuroblasts (NBs) and impressive adult dNB tumors, but not embryonic NBs. *Drosophila* embryonic and larval NBs are not equivalent. Thus, the data presented are not necessarily applicable to tumors of fetal origin. The manuscript should be re-written to de-emphasize the fetal connection. This would not undercut their conclusions.

The timing of MARCM clone induction should be included in the figure legends in Figure 1 supplements and Figure 2 and associated main text – this is important to the authors' conclusion that early-stage larval NBs are transformation competent.

The presentation of cancer stem cells (CSCs) in the manuscript is problematic, and gives specialists in CSC biology the impression that the authors do not understand the CSC hypothesis. A CSC is not a tumor cell derived from a tissue-specific stem cell. CSCs are functionally defined by their ability to recapitulate tumors in serial transplantation and limiting dilution assays. However, the authors do not present data that their Chinmo^+^ cells can recapitulate tumors upon serial transplantation (and they could do this). The issue of how their data address the CSC hypothesis is best left for the Discussion, in which they can introduce and define CSCs and compare/contrast human CSCs with their *Drosophila* tumor models, and use of the term "CSC" in the Results section should be replaced with the term "dNB" or "CIL".

The authors make a point that Chinmo^-^ cells occur in their adult pros- tumors. These Chinmo- cells may be semi-differentiated and non-tumorigenic, as the authors speculate, or they may be actively cycling NBs (transcription factors, like Dpn, are downregulated in the nucleus in M phase NBs), or something else. If the authors want to support the assertions that Chinmo^-^ cells in their pros- tumors have limited proliferation potential and that Chinmo+ cells are analogous to CSCs, then they need to do additional experiments – staining for markers (cycle, etc.), serial transplantation, and lineage tracing to show that only the Chinmo^+^ component of their tumors can recapitulate tumor growth. Should they choose to back away from their assertion that Chinmo^+^ cells are CSCs, then resolving the biology of Chinmo^-^ tumor cells is beyond the scope.

The authors compare their model to human neuroblastoma. In humans, neuroblastoma is not a primary CNS tumor, but is a tumor of the peripheral nervous system and/or adrenal glands derived from cells of neural crest origin. The authors should indicate that they know this. Whether the human neural crest cells and/or their transformed counterparts are homologous to the *Drosophila* NBs/dNBs is not known. The idea that 'neuroblastomas' are primary human brain tumors is a common misconception outside of the field of neuro-oncology. If the authors want to compare their results to data from pediatric primary CNS tumors, there are more relevant human primary brain tumors, and the Discussion should be re-written with this in mind.

Figure 7 contains too many different experiments. This figure would better support the author's conclusions if the data from the sterol-free diet were moved into a supplementary figure and if the figure included nice images of the early-L2-induced pros- clones fed on a normal diet (or are we supposed to consider the MARCM clones in Figure 1 supplements?). There should also be control stains showing Lin-28 and Imp expression at 18C as a comparison for the 29C late switch panels – this way, we can clearly see that prosRNAi induction in svp-/- cells accounts for ectopic Lin-28 and Imp expression.

Other deficits are a lack (1) of the characterization of the role of Chinmo alone (not in tumors) in NB/neural lineages and (2) data to support that Chinmo regulates imp and lin-28 in WT as well as in tumors. Could the authors also define their use of "malignant" and "benign" transformation? Do they assess metastasis?

The points described below are intended to make the authors' conclusions stronger.

1) Prior reports have shown that Chinmo is expressed in the early-born offspring of NBs (Maurange Cell 2008; Zhu Cell 2006) but in this study, Chinmo seems to be expressed in the NB itself (Figure 6). Could the authors please clarify these results in the text? Also, the first section of the Results states "Because the malignant potential of dNBs has only been assessed via transplantation experiments (Caussinus and Gonzalez, 2005), which could per se promote cancer progression, we first investigated whether dNBs induced by the loss of Prospero (Pros) acquire malignant properties when left in their natural environment. dNBs were induced in the larval VNC[…]" This set of experiments seems unnecessary as the Gonzalez lab has shown that transplantation of wild type brain tissue does not yield tumors, i.e. transplantation is not sufficient for malignancy. In addition, the loss of Prospero has previously been shown to give rise to tumors in situ. See for example, Januschke and Gonzalez' review (Oncogene. 2008 27,6994-7002) "Larval brain tissue mutant for prospero or brat develop massive malignant tumors in allograft culture (Caussinus and Gonzalez, 2005) and displays significant overgrowth in situ (Bello et al., 2006; Betschinger et al., 2006; Choksi et al., 2006; Lee et al., 2006c)".

2) The effects of loss of chinmo alone (i.e., without concomitant loss of svp or pros or brat) is not explored in this paper, and such data would really strengthen the paper.

3) Chinmo is proposed to counteract neuronal differentiation in pupae and sustain proliferation, yet prospero mutant clones in the adult contain both Chinmo positive and negative cells. Can the authors suggest how Chinmo positive cells give rise to Chinmo negative cells? Does Chinmo turn over rapidly or is there a negative regulator, which is overcome by Imp? Moreover, the authors propose that Chinmo^+^ dNBs fulfill criteria for cancer stem cells. Can this be demonstrated in a metastasis assay?

4) The effects of gain of chinmo alone (chinmo Flp-out clones, see Figure 3) needs better characterization and quantification (like that for pros RNAi Flp-out and brat MARCM in Figure 3).

5) In Figure 4, data seems to be missing and the legend is incomplete. There are no wild-type control panels to show Mira/Chinmo/Lin-28/Imp expression in normal VNCs. The stains shown in Figure 4 are confusing – yellow dashed lines are shown as though clones are presented, but not explained. In the late L3 pictures, many cells outside of the yellow lines express Lin-28, Chinmo, and Imp – what are those cells?

6) The authors do not address the role of the loss of lin-28 in their tumor model. These data would be helpful and could be obtained by a mir-sponge.

7) Most of the images are from the VNC but the authors state that their results are applicable to the central brain as well. It would be helpful to clarify in the text and each figure whether the image in from VNC or central brain. It would also be helpful to ensure that there are representative images and quantification of central brain tumors.

8) A better characterization of what Chinmo does in wild-type (non-tumors) would be helpful. Does it regulate imp and lin-28 in WT or only in pros, svp, brat mutant tumors?

9) Regarding the regulation of other target genes by Imp that determine the oncogenicity of Chinmo. Is there evidence to back up this claim?

10) The relationship of Chinmo to the temporal series is not clear to me, and it would be helpful to clarify these relationships in the text.

11) The authors describe the translational effect of Lin-28 on Chinmo and Imp, "in the tumorigenic context". Why is this effect only observed in tumor conditions? What are the other factors required?

[Editors' note: further revisions were requested prior to acceptance, as described below.]

Thank you for resubmitting your work entitled "Neural stem cell-encoded temporal patterning delineates an early window of malignant susceptibility in *Drosophila*" for further consideration at *eLife*. Your revised article has been favorably evaluated by K VijayRaghavan (Senior editor), a Reviewing editor, Hugo Bellen, and two reviewers.

The manuscript has been improved but there are some remaining issues that need to be addressed before acceptance, as outlined below. Please make the requested textual changes and corrections as suggested by the reviewers.

Reviewer #1:

In this revised manuscript, the authors have satisfied most of the experimental issues that I raised in my original critique. However, I have two related concerns about the Discussion that arise from an inaccurate comparison with "oncogene addition" and the claim that chinmo is the functional ortholog of Nmyc. Regarding oncogene addiction, this idea was first put forth more than a decade ago (see Torti 2011 EMBO Mol Med 3, 623-636), and one key component of this theory is that when the oncogene is removed/silenced in the tumor cell, that cell dies. This is not observed when chinmo is removed from the tumors. They tumors are smaller when chinmo is removed (because they proliferate less) but there are no data to suggest that the dNb lacking chinmo dies. Therefore, their data are more consistent with chinmo being a cancer stem cell gene and not an addictive oncogene. I strongly suggest that this part of the Discussion is edited.

The last two pages of the Discussion provide a lengthy conjecture on the possibility that chinmo is a functional ortholog of Nmyc. I have strong reservations about this entire section and find it naïve. First, chinmo and Nmyc have no structural/sequence similarity and they don't share the same domain structure. So this basis for their claim is not compelling. Most of the argument is based upon the observation that many genes involved in ribosome biogenesis are upregulated in dNbs, and this serves as the tenuous link between chinmo and Nmyc. In flies, myc (dm) upregulates ribosome biogenesis by upregulating Tif-1A (Grewal Nat Cell Biol 2003). Does chinmo also upregulate Tif-1A? Does chinmo upregulate myc/dm? The authors need to provide compelling data to support their argument or they need to tone down the speculation.

Reviewer #3:

Over-all the manuscript is very much improved, and the emphasis on developmental timing of transformation and the molecules that underpin transformation make this a more novel study that the previous version. Moving the comparisons to human pediatric tumors into the Discussion also vastly improved the flow and readability. The inclusion of additional experiments, particularly the transplant assay, improve the over-all quality and potential impact. Thank you to the authors.

Specific comments/suggestion for revisions are listed below:

Figure 3 – The main point of this section is to show that Chinmo is required for tumor maintenance, especially in adult stages. The authors say that in the adult stages, prosRNAi; chinmoRNAi cells have exhausted their proliferation potential, but they do not show stains or assays for proliferation to support this. Interestingly, they do show phospho-histone H3 stains of both genotypes, but only for L3 larval stages, and, at, L3, there is only a slight and not statistically significant difference in phospho-histone staining at this stage, which does not support their conclusions regarding reduced proliferative potential of prosRNAi; chinmoRNAi cells. It would be better to show phospho-histone staining of prosRNAi; chinmoRNAi cells vs. prosRNAi cells in the adult here.

Figure 4—figure supplement 2, seems to be a mix of supplementary data for Figure 3 and 4. Panels A and B are related to Figure 3 and are related to proliferation (phosho-histone staining and mitotic index of Chinmo^+^ vs. Chinmo^-^ pros RNAi cells). Panels A and B should be moved out and referred somewhere in lines 190-200. Or are they mislabeled?

In the last paragraph of the subsection “Chinmo boosts protein biosynthesis and expression of the mRNA-binding proteins Imp and Lin-28”: the authors say that Figure 4 shows wild-type Imp and lin-28 staining, but I see no wild-type brain shown. I assume that they are referring to tissue surrounding the Mira-positive cells outlined in yellow in the L3 and in the insets with asterisks, which both shown cells in poxn>prosRNAi VNCs. These are mutant brains, not wild-type brains, and are not adequate wild-type controls. No elav staining is shown to indicate that Imp-Lin-28-positive cells outside of the yellow lines are actually neurons – these cells have large nuclei consistent with an NB-like identity, rather than a neuron-identity. They should show true wild-type control NBs to support their contention that Imp and lin-28 are not expressed in normal L3 NBs (also, see below re expression in L2).

In the subsection “Lin-28 boosts Chinmo expression within tumors”: The authors state that overexpression of lin-28 in pros- cells increases the number of Chinmo^+^ Imp^+^ cells, but they only quantify the Chinmo^+^ cells in Figure 6. They show an increased number of Imp^+^ Chinmo^+^ cells in a single representative sample in Figure 6—figure supplement 2, but there is no quantification. Quantification of the effect on Imp^+^ cells is needed to support the stated conclusion.

In the subsection “Lin-28 boosts Chinmo expression within tumors”: the authors mention that their results regarding lin-28 in dNBs are consistent with publications regarding lin-28 function in human neuroblastoma. The way the sentence is worded gives the reader the impression that the authors consider human neuroblastoma evolutionarily homologous to their fly tumors, but they are not, and the authors do not otherwise show that they are. The authors are advised to remove this assertion.

In the subsection “The temporal series silences *chinmo, Imp* and *lin-28* in late NBs to limit their mitotic potential”: Here the authors state that Imp and Lin-28 are co-expressed with Chinmo is "early larval NBs and their progeny in the VNC," but they do not specifically state at which stage. This is confusing because in previous lines (subsection “Chinmo boosts protein biosynthesis and expression of the mRNA-binding proteins Imp and Lin-28”, last paragraph) they state that in L3 larval, Imp and Lin-28 are not expressed in NBs. The subsection “The temporal series silences *chinmo, Imp* and *lin-28* in late NBs to limit their mitotic potential” should be revised to more accurately state that timing of Imp and Lin-28 expression in L2 NBs and not in late L3 in normal brains. The way the data in the last paragraph of the subsection “Chinmo boosts protein biosynthesis and expression of the mRNA-binding proteins Imp and Lin-28” are presented leaves the reader with the impression that Imp and Lin-28 expression are ectopically acquired in the L3 and adult tumor cells, rather than inappropriately maintained past L2. Also, if the timing of Imp and lin-28 expression in wild-type is so crucial to understanding the tumor phenotypes, then proper wild-type L3 control stains should be presented as mentioned above.

Figure 8 – Note more clearly in main text or figure legend that, in D, the GFP and Mira staining show widespread proliferative dNBs in the CNS – the image show in the L1/L2 panels is low magnification and the green and red staining appear to be so widespread that they look like background. Or maybe a higher magnificent image of part of the brain could be shown?

In the first paragraph of the subsection “An archaic model for human neural tumors with early developmental origins?”: It is not an established fact that medulloblastomas (MB), PNETs, and other neural tumors arise in the fetal or embryonic stages. Plenty of data exist to contradict this assertion, and the citations the authors use do not support this assertion (Kaatch is an epidemology review, and Vogelstein is a review about cancer genomics in general). For example, several experimental mouse models of MB show that tumors can arise post-natally, and the peak ages of incidence for MB is 3-5 and 8-9 years of age (diagnoses of children under 1 year of age are uncommon). The origins of these tumors is very much an active area of investigation. The similarities between the tumor cells and fetal/embryonic stem cells should not be construed to mean that these tumors therefore arise from fetal/embryonic cell types. Indeed, the term "embryonal" is not really meant to refer to these tumors' origins, but rather is a pathology term that refers to their common histological and immunohistochemical features: cells within these tumors show specific similarities to each other and to immature cells in their tissues of origin. Again, the authors misconstrue medical terminology and basic science terminology. This section of the Discussion should be re-written to de-emphasize that such tumors obligately arise from embryonic/fetal stages, and should be written to be more broadly inclusive of childhood tumors, which are thought to arise both pre- and post-natally.

An appropriate citation is:

Childhood Central Nervous System Embryonal Tumors Treatment (PDQ)

Health Professional Version. PDQ Pediatric Treatment Editorial Board. March 16, 2016.

---

## [Author Response]

Summary:

Early neural development is tightly controlled by a series of transcription factors that are expressed in a temporal fashion. In this study, the authors test the hypothesis that malignant transformation of neural progenitor cells occurs by disrupting the regulation of the temporal pattern of transcription factors that drive neuronal development. The authors find that early stage postembryonic neural progenitor cells, but not late stage neural progenitor cells, are prone to malignant transformation. Moreover, the authors present data that support a conserved role for Chinmo, Lin-28 and Imp for competency of progenitor cells to undergo malignant transformation. This is a novel and elegant study of broad interest, but there is room for improvement. The results are somewhat obscured by the fact that the paper is rather long and the genetics quite complex. The manuscript would benefit from shortening and bringing out the key results more clearly.

In line with the recommendation of the editor and reviewers, we have tried to shorten the manuscript while highlighting and reinforcing our key results. This has involved the addition of several new experiments that can be seen in Figure 2—figure supplement 1(stainings of tumors with anti-Pdm and anti-Svp), Figure 3 (Transplantation experiments), Figure 6—figure supplement 1 (role of Lin-28 in tumor growth), and Figure 7 (role of Chinmo in NBs during development). We have also modified several figures, adding additional quantifications and controls to satisfy reviewers’ requests (e.g. Figure 1, Figure 1—figure supplement 2, Figure 2, Figure 3—figure supplement 2, Figure 4—figure supplement 2) and decided to remove several experiments (ex-Figure 3 and ex-Figure 5) that we think were not essential to convey the main findings and were sometimes shifting the focus towards issues beyond the scope of the study. We believe this current manuscript represents a clarified and improved version.

The Abstract/Introduction/Discussion focuses on fetal origins of human pediatric neural tumors. However, the Drosophila data presented is mainly from postembryonic larval neuroblasts (NBs) and impressive adult dNB tumors, but not embryonic NBs. Drosophila embryonic and larval NBs are not equivalent. Thus, the data presented are not necessarily applicable to tumors of fetal origin. The manuscript should be re-written to de-emphasize the fetal connection. This would not undercut their conclusions.

First, we would like to clarify that adult malignant tumors in our manuscript have always been induced during an early window of larval development (from L1 to early L3). Therefore, they are not “adult” tumors but “developmental” tumors. In order to clarify this point in the text, we have replaced the term “adult tumors” by more explicit terms, such as “tumors that persist in adults”.

Concerning the question of the possible correspondence of embryonic and larval stages in insects to fetal stages in mammals, it is actually difficult to determine which *Drosophila* developmental stages resemble more fetal stages in mammals. Interestingly, in the mouse cortex, both Imp1/Igf2bp1 and Lin28a are highly expressed in early fetal neural stem cells but their expression decreases from E14 and to be silenced at birth (Balzer et al. 2010, Development; Nishino et al. 2013, *eLife*; Yang et al. 2015, Development). This is reminiscent to our observation that Imp and Lin-28 are highly expressed in early larval NBs and are silenced during L3 stages. Therefore, in terms of neural development, the larval period during which Imp and Lin-28 are expressed in NBs could be equivalent to embryonic/fetal stages in mice when Imp1 and Lin28a are expressed.

Moreover, late stages of embryogenesis and early fetal stages in mammals are typically characterized by high rates of cell proliferation and organismal growth that slow down towards birth. In *Drosophila*, there is no organismal growth during embryogenesis, but larval stages exhibit high rates of both cell proliferation and organismal growth (Church and Robertson 1966, Journal of Experimental Zoology; Tenessen and Thummel 2011, Current Biology). Precisely, body mass increases exponentially from late L1 to mid-L3 stages, when larvae are subjected to ecdysone pulses in order to start metamorphosis. Therefore, the first 2/3 of larval stages in *Drosophila* are very similar to late embryonic and fetal stages in mammals in that they constitute an early period of development with very high levels of cell proliferation and growth and low levels of differentiation.

In our opinion, these data argue that the regulation of tissue growth during early larval stages in *Drosophila* and late embryonic/early fetal stages in mammals may be very similar, in particular concerning the nervous system. We therefore think it is important to keep the analogy between this model of neural tumor in *Drosophila* and tumors with early developmental (embryonic or fetal) origins in humans, like embryonal neural tumors. To clarify this issue, we have now emphasized the similarities between larval NBs in *Drosophila* and fetal NSCs in mammals in the Discussion (subsection “Temporal regulation of NSC mitotic potential during development”, second paragraph), and we have removed the term “fetal” from the Abstract and tried to tone down this aspect in the Introduction.

The timing of MARCM clone induction should be included in the figure legends in Figure 1 supplements and Figure 2 and associated main text – this is important to the authors' conclusion that early-stage larval NBs are transformation competent.

We have now added the timing of MARCM clone induction (L1/L2 or mid-L3) in all appropriate figures and in the associated main text.

The presentation of cancer stem cells (CSCs) in the manuscript is problematic, and gives specialists in CSC biology the impression that the authors do not understand the CSC hypothesis. A CSC is not a tumor cell derived from a tissue-specific stem cell. CSCs are functionally defined by their ability to recapitulate tumors in serial transplantation and limiting dilution assays. However, the authors do not present data that their Chinmo^+^ cells can recapitulate tumors upon serial transplantation (and they could do this). The issue of how their data address the CSC hypothesis is best left for the Discussion, in which they can introduce and define CSCs and compare/contrast human CSCs with their Drosophila tumor models, and use of the term "CSC" in the Results section should be replaced with the term "dNB" or "CIL".

We thank the reviewers for alerting us on this important point and agree that the work here is too preliminary to unambiguously demonstrate that Chinmo^+^ dNBs are CSCs. There is theoretically the possibility that Chinmo^+^ dNBs act as a transient amplifying population of cells necessary for tumor growth but with a limited self-renewal potential, while the CSC population would be a rare population of Chinmo^-^ cells with low mitotic activity. As suggested by the reviewer, we replaced the term “CSC” by the term “Chinmo^+^ dNBs” throughout the Results section, and discuss the possibility of CSCs in the Discussion (subsection “An oncogenic module defining growth-sustaining cells in neural tumors with early developmental origins”, last paragraph).

The authors make a point that Chinmo^-^ cells occur in their adult pros- tumors. These Chinmo- cells may be semi-differentiated and non-tumorigenic, as the authors speculate, or they may be actively cycling NBs (transcription factors, like Dpn, are downregulated in the nucleus in M phase NBs), or something else. If the authors want to support the assertions that Chinmo^-^ cells in their pros- tumors have limited proliferation potential and that Chinmo+ cells are analogous to CSCs, then they need to do additional experiments – staining for markers (cycle, etc.), serial transplantation, and lineage tracing to show that only the Chinmo^+^ component of their tumors can recapitulate tumor growth. Should they choose to back away from their assertion that Chinmo^+^ cells are CSCs, then resolving the biology of Chinmo^-^ tumor cells is beyond the scope.

We have now confirmed that at least a large fraction of Chinmo^-^ cells in tumors are actively cycling dNBs by showing a co-staining of PH3 (to show mitotic activity), Miranda (to show NB identity), and Chinmo (new Figure 4—figure supplement 2). Our quantitative analysis shows that the fraction of mitotic Chinmo^-^ NBs is lower than for Chinmo^+^ NBs (Figure 4—figure supplement 2). This can be explained by slow cycling chinmo^-^ NBs, or by the existence of a sub-population of chinmo^-^ NBs that have undergone cell-cycle exit. We agree that we cannot unambiguously demonstrate that Chinmo^+^ cells are CSCs and that Chinmo^-^ cells have a limited proliferation potential without performing additional lineage tracing and transplantation experiments. Therefore, we have indeed chosen to back away from our initial assertion and now only discuss these possibilities in the Discussion (subsection “An oncogenic module defining growth-sustaining cells in neural tumors with early developmental origins”, last paragraph). These questions will be thoroughly addressed in another study. We have instead chosen to reinforce our finding that Chinmo is required and sufficient for sustained tumor growth by performing transplantation experiments (new Figure 3).

The authors compare their model to human neuroblastoma. In humans, neuroblastoma is not a primary CNS tumor, but is a tumor of the peripheral nervous system and/or adrenal glands derived from cells of neural crest origin. The authors should indicate that they know this. Whether the human neural crest cells and/or their transformed counterparts are homologous to the Drosophila NBs/dNBs is not known. The idea that 'neuroblastomas' are primary human brain tumors is a common misconception outside of the field of neuro-oncology. If the authors want to compare their results to data from pediatric primary CNS tumors, there are more relevant human primary brain tumors, and the Discussion should be re-written with this in mind.

The reason we initially emphasized the resemblance of neuroblastoma with *Drosophila* NB tumors is because of the expression and role of Lin28 and Imp that has essentially been described in this kind of pediatric cancer. Whether *Drosophila* NB tumors, as an archaic model, resemble more pediatric primary CNS or PNS tumors remains to be clearly investigated. We have now modified the Discussion and added the corresponding tissue of origin for each tumor type discussed.

Figure 7 contains too many different experiments. This figure would better support the author's conclusions if the data from the sterol-free diet were moved into a supplementary figure and if the figure included nice images of the early-L2-induced pros- clones fed on a normal diet (or are we supposed to consider the MARCM clones in Figure 1 supplements?). There should also be control stains showing Lin-28 and Imp expression at 18C as a comparison for the 29C late switch panels – this way, we can clearly see that prosRNAi induction in svp-/- cells accounts for ectopic Lin-28 and Imp expression.

We have now split ex-Figure 7 (now new Figure 8 and Figure 9) and the text accordingly. New Figure 8 now concentrates on the different phenotypes generated by inducing tumors at early (L1/L2) or late (mid-L3) larval stages demonstrating that malignant Chinmo^+^ tumors can only be induced during an early window of development. The figure now contains image of early- and late- induced MARCM clones, 1 day, 3 days and 7 days after induction.

New Figure 9 now focuses on the role of the temporal transcription factor series in the regulation of this early window of malignant susceptibility. The goal of the temperature switching experiment is to demonstrate that the presence of Chinmo, Imp and Lin-28 in tumors and their ability to become malignant depend on the temporal identity of the cell of origin, and not on the developmental stage of the organism (here we block temporal progression in the *svp^-/-^* NBs without affecting developmental progression of the organism).

The 18°C control experiment aims at showing that the *pros^RNAi^*is ineffective when larvae are kept at this temperature (due to the GAL80^ts^-mediated repression). Therefore, tumors seen in the “late switch to 29°C” experiments are really induced by the late temperature switch, and were not preexisting at earlier stages. As shown, the *pros^RNAi^*is ineffective at 18°C, this experiment is therefore similar to inducing *svp^-/-^*MARCM clones. We have shown that these clones do retain Chinmo, Imp and Lin-28 expression when induced in L1/L2 and observed in adults (Figure 7). Thus, we don’t think reassessing the expression of these factors in the 18°C experiment would bring novel insights. However, we have now better explained the experiments in the main text (subsection “The temporal identity of a cell determines its malignant susceptibility”) and in the Materials and methods (subsection “Fly lines”).

Other deficits are a lack (1) of the characterization of the role of Chinmo alone (not in tumors) in NB/neural lineages and (2) data to support that Chinmo regulates imp and lin-28 in WT as well as in tumors.

We have now looked more carefully at the role of Chinmo during larval neurogenesis by inducing loss-of-function clones. Our findings suggest that Chinmo is dispensable for NB mitotic activity during development since chinmo^-/-^ clones contained the same number of cells as *wt* clones (new Figure 7) at the end of larval stages. This finding is corroborated by the fact that *chinmo^RNAi^, svp^-/-^*NBs, blocked in an early temporal identity, continued to divide in adults (Figure 7). Thus, the growth of NB lineages during development and tumorigenesis does not exhibit the same dependency on Chinmo. This reveals a tumor-specific Chinmo addiction. This point has now been added in the Results (subsection “The temporal series silences *chinmo, Imp* and *lin-28* in late NBs to limit their mitotic potential”, first paragraph) and Discussion (subsection “Temporal regulation of NSC mitotic potential during development”).

Intriguingly, *chinmo^RNAi^, svp^-/-^*NBs maintained Imp expression showing that, in contrast to dNBs (Figure 4—figure supplement 1), Imp expression in NBs with an early temporal identity does not rely on Chinmo (Figure 7). Thus, Chinmo and Imp in progenitors are not subordinated to similar cross-regulatory interactions whether in the developmental or in the tumorigenic context (subsection “Temporal regulation of NSC mitotic potential during development”, last paragraph).

However, *Imp^RNAi^, chinmo^RNAi^, svp^-/-^* NBs did not persist in adults (Figure 7). Thus, the presence of the Chinmo/Imp module is required to sustain the mitotic activity of NBs blocked in an early temporal identity. It is therefore important that the temporal series switches off the Chinmo/Imp module during late larval stages to limit their proliferation potential and ensure that NBs become competent to undergo terminal differentiation during metamorphosis.

Could the authors also define their use of "malignant" and "benign" transformation? Do they assess metastasis?

Malignant tumors, as we define them, are the ones that resist differentiation during metamorphosis and continue growing at a fast rate in adults. This is assessed by their ability to cover the whole VNC (or brain in case of *brat^-/-^* clones) after about 5 days. We have also observed that these tumors after 5 days invade adjacent tissues, mainly by following axonal bundles emanating from the CNS (not shown). However, we have not systematically assessed this property in our assays.

In contrast, benign tumors are the ones that have a limited ability to persist in adults as they tend to be more efficiently eliminated during metamorphosis upon differentiation cues. When they persist in adults, they do not grow, or grow at a very low rate such that the VNC, or brain is never covered after 5 days.

We have tried to clarify this issue, in the Results section by adding:

“Therefore, NB tumors induced by the loss of Pros during early larval stages and maintained in their natural environment resist differentiation cues operating during metamorphosis, and invariably acquire an unlimited growth potential as well as invasive properties. As such, we define them as malignant tumors.”and “In contrast, dNBs induced by the dedifferentiation of late-born CIL-negative GMCs cannot reactivate Chinmo expression, possess a limited proliferation potential and undergo terminal differentiation during metamorphosis like wt NBs, or persist as benign, slow-growing, tumors in adults.”

The points described below are intended to make the authors' conclusions stronger.

1) Prior reports have shown that Chinmo is expressed in the early-born offspring of NBs (Maurange Cell 2008; Zhu Cell 2006) but in this study, Chinmo seems to be expressed in the NB itself (Figure 6). Could the authors please clarify these results in the text?

We have clarified this point in the text: “We then concentrated on Chinmo, a transcription factor known to label a sub-population of early-born neurons in type-I lineages in larvae (Maurange et al., 2008; Zhu et al., 2006), Figure 2). […] Therefore, Chinmo is a marker of early temporal identity and its window of expression in NBs is terminated in early L3 by progression of the temporal transcription factor series (Figure 2).”

Also, the first section of the Results states "Because the malignant potential of dNBs has only been assessed via transplantation experiments (Caussinus and Gonzalez, 2005), which could per se promote cancer progression, we first investigated whether dNBs induced by the loss of Prospero (Pros) acquire malignant properties when left in their natural environment. dNBs were induced in the larval VNC…" This set of experiments seems unnecessary as the Gonzalez lab has shown that transplantation of wild type brain tissue does not yield tumors, i.e. transplantation is not sufficient for malignancy. In addition, the loss of Prospero has previously been shown to give rise to tumors in situ. See for example, Januschke and Gonzalez' review (Oncogene. 2008 27,6994-7002) "Larval brain tissue mutant for prospero or brat develop massive malignant tumors in allograft culture (Caussinus and Gonzalez, 2005) and displays significant overgrowth in situ (Bello et al., 2006; Betschinger et al., 2006; Choksi et al., 2006; Lee et al., 2006c)".

In the first manuscript, it was not our intention to imply that transplantation might be sufficient for malignancy but instead that the process of transplantation, that involves cutting the tumor in pieces and injecting it in the abdominal cavity of adult flies, could favor the transformation of tumor cells (for example by favoring cell dissemination in the hemolymph and the apparition of secondary tumors at distant sites, or by activating growth pathways triggered by wounds). Therefore, the possible contribution of transplantation to malignancy remains unclear. It therefore seemed important to us to design an assay in which early developmental tumors can be precisely tracked, to investigate how their differentiating, proliferative and invasive properties may evolve over several days/weeks, throughout development (in the case of transplantation, usually larval tumors are directly transplanted in adults) and in their natural environment with no mechanic interventions.

Our assay using *pox>pros^RNAi^* tumors induced from a defined subset of 6 NBs localized in the VNC has allowed us to show that early induced tumors undergo a period of regression during metamorphosis, demonstrating that not all dNBs have the same ability to resist differentiation cues operating in pupae (resistance to differentiation is a hallmark of cancer cells). Moreover, intact tumors never stop growing (even more than three weeks after initiation) and can invade surrounding tissues in adults. Thus, the malignant potential of tumors is not triggered by transplantation, although transplantation could still accelerate the acquisition of novel cancer hallmarks. We believe that our assay nicely complement the transplantation assay in the quest to uncover the natural events that promote early phases of malignant transformation.

We have modified the text in the first paragraph of the Results section to better explain the advantages of the *pox>pros^RNAi^* assay, and have also added new transplantation experiments to support our data concerning untransplanted tumors (new Figure 3).

2) The effects of loss of chinmo alone (i.e., without concomitant loss of svp or pros or brat) is not explored in this paper, and such data would really strengthen the paper.

We have now investigated this issue that is discussed in the last response to the Main issues.

3) Chinmo is proposed to counteract neuronal differentiation in pupae and sustain proliferation, yet prospero mutant clones in the adult contain both Chinmo positive and negative cells. Can the authors suggest how Chinmo positive cells give rise to Chinmo negative cells? Does Chinmo turn over rapidly or is there a negative regulator, which is overcome by Imp? Moreover, the authors propose that Chinmo^+^ dNBs fulfill criteria for cancer stem cells. Can this be demonstrated in a metastasis assay?

For the moment, we do not know how the self-renewal of Chinmo^+^ dNBs or their probable transition to a Chinmo^-^ state is regulated. This is a very interesting question and is something we are actively working on. We think however that it goes beyond the scope of the current manuscript. As mentioned in response to the third and fourth Main issue, we agree that we have not formally demonstrated that Chinmo^+^ cells are at the top of the tumor hierarchy and therefore have removed allusions to CSCs in the Results section. We also recognize that our lineage analysis is too preliminary to support the hypothesis that Chinmo^+^ cells are CSCs and have therefore removed this Figure (ex Figure 3) from the new version of the manuscript. This hypothesis is now only mentioned in the Discussion (subsection “Temporal regulation of NSC mitotic potential during development”, last paragraph).

4) The effects of gain of chinmo alone (chinmo Flp-out clones, see Figure 3) needs better characterization and quantification (like that for pros RNAi Flp-out and brat MARCM in Figure 3).

Because adult flies containing chinmo Flp-out clones die rapidly (most of them do not actually emerge from the pupa), it is not possible to quantify their growth capacity over several days as we have done from *pros^RNAi^*and *brat^-/-^* MARCM clones. In order to circumvent this issue, we have now transplanted larval CNSs with all NBs and their GMCs overexpressing chinmo (new Figure 3). We could show that in 26% of the cases, it is sufficient to induce tumors filling the whole abdominal cavity after 7 days. This nicely supports our initial statement that Chinmo is oncogenic and sufficient to induce sustained tumorigenic growth (subsection “Chinmo promotes resistance to differentiation and is necessary to sustain tumor growth”, last paragraph).

5) In Figure 4, data seems to be missing and the legend is incomplete. There are no wild-type control panels to show Mira/Chinmo/Lin-28/Imp expression in normal VNCs. The stains shown in Figure 4 are confusing – yellow dashed lines are shown as though clones are presented, but not explained. In the late L3 pictures, many cells outside of the yellow lines express Lin-28, Chinmo, and Imp – what are those cells?

We have now added in Figure 4 an inset depicting a normal VNC NB and its progeny in late L3 stained with Chinmo, Imp and Lin-28. Expression of Chinmo, Lin28, Imp in VNCs throughout larval development is presented in more details in Figure 7. We have also completed the legend and explained that the cells that express Chinmo, Imp and Lin28 outside the clones in late L3 larvae are early-born neurons.

6) The authors do not address the role of the loss of lin-28 in their tumor model. These data would be helpful and could be obtained by a mir-sponge.

We have now shown, by generating *brat^-/-^* MARCM clones in a *lin-28* homozygous mutant background or by generating *pros^RNAi^ lin-28^-/-^* MARCM clones, that Lin-28 is dispensable for tumor growth and expression of Chinmo and Imp (new Figure 6—figure supplement 1). Based on our results showing that high levels of Lin-28 can enhance the levels of Chinmo and Imp in tumors (Figure 6), we have defined *lin-28* as being able to boost Chinmo/Imp oncogenic module, and not as being a core component of the module (end of subsection “Lin-28 boosts Chinmo expression within tumors”).

7) Most of the images are from the VNC but the authors state that their results are applicable to the central brain as well. It would be helpful to clarify in the text and each figure whether the image in from VNC or central brain. It would also be helpful to ensure that there are representative images and quantification of central brain tumors.

The identity of the examined tissues (VNCs or brains) has been systematically added to all images. We have quantified the volume of pros^RNAi^tumors in the brain (new Figure 3—figure supplement 2) as well as in brat^-/-^ tumors in the brain (Figure 6—figure supplement 1 and Figure 3).

8) A better characterization of what Chinmo does in wild-type (non-tumors) would be helpful. Does it regulate imp and lin-28 in WT or only in pros, svp, brat mutant tumors?

See the last response to the Main issues.

9) Regarding the regulation of other target genes by Imp that determine the oncogenicity of Chinmo. Is there evidence to back up this claim?

Our evidence was that expressing Chinmo in *Imp^-/-^* clones (using the transgene that lack the 5’UTR, and therefore does not depend on Imp for its translation) does not induce NB amplification, while expressing Chinmo in *wt* NBs does (ex-Figure 5 in old version). Therefore, Imp must regulate other genes to promote the oncogenic activity of Chinmo. However, as advised by the editor, we have decided to remove this set of experiments, which does not bring essential information to the main messages of the study, in order to keep the study more focused.

10) The relationship of Chinmo to the temporal series is not clear to me, and it would be helpful to clarify these relationships in the text.

We had previously shown that Chinmo in neurons is downstream to the temporal series (Maurange et al. 2008, Cell). We now extent this relationship to NBs as they are retained in late NBs that are temporally blocked (as shown in *svp^-/-^* clones for example – Figure 2) and to Imp and Lin28 (Figure 7). We have tried to clarify this relationship in the Introduction as well as the Results section entitled: “A subset of dNBs aberrantly maintains the early transcription factor Chinmo”. More details are also given in the Results section entitled: “The temporal series silences *chinmo, Imp* and *lin-28* in late NBs to limit their mitotic potential”.

11) The authors describe the translational effect of Lin-28 on Chinmo and Imp, "in the tumorigenic context". Why is this effect only observed in tumor conditions? What are the other factors required?

It is true that during development, mis-expression of Lin-28 and Imp is not sufficient to promote the persistence of Chinmo beyond its normal temporal window of expression. We can only speculate that there may be some repressors of Chinmo in late NBs that counteract Lin-28 activity. Moreover, as shown in Figure 4—figure supplement 3 and Figure 7, *chinmo, Imp*, and probably *lin-28* are not subordinated to similar cross-regulatory interactions whether in the developmental or tumorigenic context. This is probably due to the fact that the Chinmo/Imp/lin-28 module has been coopted in dNBs but the upstream regulatory mechanisms might be different than in normal NBs. However, we think that uncovering the molecular basis of this interesting question goes beyond the scope of the current manuscript.

[Editors' note: further revisions were requested prior to acceptance, as described below.]

The manuscript has been improved but there are some remaining issues that need to be addressed before acceptance, as outlined below. Please make the requested textual changes and corrections as suggested by the reviewers.

Reviewer #1:

In this revised manuscript, the authors have satisfied most of the experimental issues that I raised in my original critique. However, I have two related concerns about the Discussion that arise from an inaccurate comparison with "oncogene addition" and the claim that chinmo is the functional ortholog of Nmyc. Regarding oncogene addiction, this idea was first put forth more than a decade ago (see Torti 2011 EMBO Mol Med 3, 623-636), and one key component of this theory is that when the oncogene is removed/silenced in the tumor cell, that cell dies. This is not observed when chinmo is removed from the tumors. They tumors are smaller when chinmo is removed (because they proliferate less) but there are no data to suggest that the dNb lacking chinmo dies. Therefore, their data are more consistent with chinmo being a cancer stem cell gene and not an addictive oncogene. I strongly suggest that this part of the Discussion is edited.

We agree that we have not carefully investigated whether loss of Chinmo triggers enhanced apoptosis within the tumor. We have therefore removed the comparison with oncogene addiction.

The last two pages of the Discussion provide a lengthy conjecture on the possibility that chinmo is a functional ortholog of Nmyc. I have strong reservations about this entire section and find it naïve. First, chinmo and Nmyc have no structural/sequence similarity and they don't share the same domain structure. So this basis for their claim is not compelling. Most of the argument is based upon the observation that many genes involved in ribosome biogenesis are upregulated in dNbs, and this serves as the tenuous link between chinmo and Nmyc. In flies, myc (dm) upregulates ribosome biogenesis by upregulating Tif-1A (Grewal Nat Cell Biol 2003). Does chinmo also upregulate Tif-1A? Does chinmo upregulate myc/dm? The authors need to provide compelling data to support their argument or they need to tone down the speculation.

As suggested, we have toned down the speculative discussion about Chinmo being a functional analog of MYCN. Instead, we propose that Chinmo/Imp/Lin-28 in insects may compose an oncogenic module with similar functions during development and tumorigenesis as MYCN/IMP/LIN28 in mammals. This remains speculative but not completely unfounded based on recent evidences about their temporal and spatial regulation, cross-regulatory interactions and downstream transcriptional landscape.

*Reviewer #3:*

*Specific comments/suggestion for revisions are listed below:*

Figure 3 – The main point of this section is to show that Chinmo is required for tumor maintenance, especially in adult stages. The authors say that in the adult stages, prosRNAi; chinmoRNAi cells have exhausted their proliferation potential, but they do not show stains or assays for proliferation to support this. Interestingly, they do show phospho-histone H3 stains of both genotypes, but only for L3 larval stages, and, at, L3, there is only a slight and not statistically significant difference in phospho-histone staining at this stage, which does not support their conclusions regarding reduced proliferative potential of prosRNAi; chinmoRNAi cells. It would be better to show phospho-histone staining of prosRNAi; chinmoRNAi cells vs. prosRNAi cells in the adult here.

We did show that already in larvae, the mitotic rate in *pros^RNAi^; chinmo^RNAi^* tumors is significantly lower than in *pros^RNAi^* tumors (Figure 3—figure supplement 1). In addition, we have now added some new results showing that the mitotic rate of dNBs is lower in flp-out *pros^RNAi^, chinmo^RNAi^*clones compared to *pros^RNAi^*clones, both in larvae and adults. Moreover, we find that the mitotic index of *pros^RNAi^, chinmo^RNAi^* clones decreases further in adults compared to larvae, while remaining constant in *pros^RNAi^*clones. Together, these new results support a progressive exhaustion of proliferation potential in tumors lacking Chinmo. This new data has now been incorporated as Figure 3.

Figure 4—figure supplement 2, seems to be a mix of supplementary data for Figure 3 and 4. Panels A and B are related to Figure 3 and are related to proliferation (phosho-histone staining and mitotic index of Chinmo^+^ vs Chinmo^-^ pros RNAi cells). Panels A and B should be moved out and referred somewhere in lines 190-200. Or are they mislabeled?

We agree with Reviewer #3 that the results for Figure 4—figure supplement 2 A-B better fits in the previous section. It nicely shows that two types of progenitors with different mitotic properties co-exist in tumors (now Figure 3—figure supplement 3). We have also added new counts that indicate the average fraction of Chinmo^+^ vs Chinmo^-^ dNBs in tumors (now Figure 2—figure supplement 5). In the future, we plan to perform detailed lineage analyses in order to precisely determine the proliferation potential of each progenitor population.

In the last paragraph of the subsection “Chinmo boosts protein biosynthesis and expression of the mRNA-binding proteins Imp and Lin-28”: the authors say that Figure 4 shows wild-type Imp and lin-28 staining, but I see no wild-type brain shown. I assume that they are referring to tissue surrounding the Mira-positive cells outlined in yellow in the L3 and in the insets with asterisks, which both shown cells in poxn>prosRNAi VNCs. These are mutant brains, not wild-type brains, and are not adequate wild-type controls. No elav staining is shown to indicate that Imp-Lin-28-positive cells outside of the yellow lines are actually neurons – these cells have large nuclei consistent with an NB-like identity, rather than a neuron-identity. They should show true wild-type control NBs to support their contention that Imp and lin-28 are not expressed in normal L3 NBs (also, see below re expression in L2).

We agree that the presentation of the immunostaining data was a bit confusing. We have now modified the text and Figure 4 such that Figure 4 only describes the expression patterns of Imp and Lin-28 in tumors, and not anymore in the wild type CNS. Here we only aim at validating the transcriptomic analysis that was performed on tumors.

Description of Imp and Lin-28 expression in the wild type CNS is now restricted to Figure 7. We have added additional data (Figure 7—figure supplement 1) to show that cells, around larval NBs, co-expressing Imp and Lin-28 during development are elav+ neurons. We have also included new stainings showing that Chinmo and Lin-28 are completely silenced in the CNS past metamorphosis, while Imp perdures in some adult neurons. We hope this clarifies the issue.

In the subsection “Lin-28 boosts Chinmo expression within tumors”: The authors state that overexpression of lin-28 in pros- cells increases the number of Chinmo+ Imp+ cells, but they only quantify the Chinmo+ cells in Figure 6. They show an increased number of Imp^+^.Chinmo^+^ cells in a single representative sample in Figure 6—figure supplement 2, but there is no quantification. Quantification of the effect on Imp+ cells is needed to support the stated conclusion.

We have not, indeed, precisely quantified Imp^+^ dNBs in tumors over-expressing Lin-28. However, we show in Figure 6—figure supplement 2 that in such tumors, all Chinmo^+^ cells are also Imp^+^. Given that the number of Chinmo^+^ cells increases, this by extension implies that the number of Imp^+^ cells increases in proportion. We have modified the text to make our assertion clearer and propose that Lin-28 over-expression in the tumorigenic context is able to favor the self-renewing capacity of Chinmo^+^/Imp^+^ dNBs at the expense of Chinmo^-^/Imp^-^ dNBs.

In the subsection “Lin-28 boosts Chinmo expression within tumors”: the authors mention that their results regarding lin-28 in dNBs are consistent with publications regarding lin-28 function in human neuroblastoma. The way the sentence is worded gives the reader the impression that the authors consider human neuroblastoma evolutionarily homologous to their fly tumors, but they are not, and the authors do not otherwise show that they are. The authors are advised to remove this assertion.

We have now removed the mention to neuroblastoma in this sentence.

In the subsection “The temporal series silences chinmo, Imp and lin-28 in late NBs to limit their mitotic potential”: Here the authors state that Imp and Lin-28 are co-expressed with Chinmo is "early larval NBs and their progeny in the VNC," but they do not specifically state at which stage. This is confusing because in previous lines (subsection “Chinmo boosts protein biosynthesis and expression of the mRNA-binding proteins Imp and Lin-28”, last paragraph) they state that in L3 larval, Imp and Lin-28 are not expressed in NBs. The subsection “The temporal series silences chinmo, Imp and lin-28 in late NBs to limit their mitotic potential” should be revised to more accurately state that timing of Imp and Lin-28 expression in L2 NBs and not in late L3 in normal brains.

The text has been modified accordingly. We have also added a Figure 7—figure supplement 1 that shows i) that Imp^+^ cells surrounding NBs in larvae are elav^+^ neurons, ii) that Lin-28 and Chinmo are respectively silenced in all CNS cells during late larval and metamorphosis respectively, while Imp persist in a vast number of neurons in the adult CNS.

The way the data in the last paragraph of the subsection “Chinmo boosts protein biosynthesis and expression of the mRNA-binding proteins Imp and Lin-28” are presented leaves the reader with the impression that Imp and Lin-28 expression are ectopically acquired in the L3 and adult tumor cells, rather than inappropriately maintained past L2.

Indeed, at this stage in the manuscript, we still don’t know whether aberrant expression of Imp, Lin28 and Chinmo in tumors is due to ectopic expression or inappropriate maintenance, this is the purpose of the last two Results sections of the manuscript to demonstrate that this is due to inappropriate maintenance (Figure 8 and Figure 9). We hope that the presentation of the data is now clearer with the modifications we have performed in Figure 4.

Also, if the timing of Imp and lin-28 expression in wild-type is so crucial to understanding the tumor phenotypes, then proper wild-type L3 control stains should be presented as mentioned above.

Done in Figure 7—figure supplement 1.

Figure 8 – Note more clearly in main text or figure legend that, in D, the GFP and Mira staining show widespread proliferative dNBs in the CNS – the image show in the L1/L2 panels is low magnification and the green and red staining appear to be so widespread that they look like background. Or maybe a higher magnificent image of part of the brain could be shown?

We have added a higher magnification that clearly shows that red and green stainings are proliferating dNBs.

In the first paragraph of the subsection “An archaic model for human neural tumors with early developmental origins?”: It is not an established fact that medulloblastomas (MB), PNETs, and other neural tumors arise in the fetal or embryonic stages. Plenty of data exist to contradict this assertion, and the citations the authors use do not support this assertion (Kaatch is an epidemology review, and Vogelstein is a review about cancer genomics in general). For example, several experimental mouse models of MB show that tumors can arise post-natally, and the peak ages of incidence for MB is 3-5 and 8-9 years of age (diagnoses of children under 1 year of age are uncommon). The origins of these tumors is very much an active area of investigation. The similarities between the tumor cells and fetal/embryonic stem cells should not be construed to mean that these tumors therefore arise from fetal/embryonic cell types. Indeed, the term "embryonal" is not really meant to refer to these tumors' origins, but rather is a pathology term that refers to their common histological and immunohistochemical features: cells within these tumors show specific similarities to each other and to immature cells in their tissues of origin. Again, the authors misconstrue medical terminology and basic science terminology. This section of the Discussion should be re-written to de-emphasize that such tumors obligately arise from embryonic/fetal stages, and should be written to be more broadly inclusive of childhood tumors, which are thought to arise both pre- and post-natally.

An appropriate citation is:

Childhood Central Nervous System Embryonal Tumors Treatment (PDQ)

Health Professional Version. PDQ Pediatric Treatment Editorial Board. March 16, 2016.

We have now corrected the misuse of the term “embryonal”, and mention the fact that the embryonic/fetal origin of tumors is only suspected for a number of pediatric cancers. We also added a recent reference showing that in mice, Smarcb1 loss can only induce AT/RTs if induced during a restricted window of embryonic development (Han et al. 2016). Because our study involves genes like Lin-28 and Imp that are mainly expressed during embryonic/fetal stages in mammals, emphasizing these types of cancers seems pertinent to us. But although, pediatric cancers appear to be initiated during specific windows of development associated with high proliferation, we agree that depending on the tissue, it may not always be restricted to embryonic/fetal stages.